# Simulating preferential soil water flow and tracer transport using the Lagrangian Soil Water and Solute Transport Model

Alexander Sternagel[1,2], Ralf Loritz[1], Wolfgang Wilcke[2], Erwin Zehe[1]

[1] Karlsruhe Institute of Technology (KIT), Institute of Water Resources and River Basin Management, Hydrology
[2] Karlsruhe Institute of Technology (KIT), Institute of Geography and Geoecology, Geomorphology and Soil Science

*Correspondence to*: Alexander Sternagel (alexander.sternagel@kit.edu)

**Abstract.** We propose an alternative model concept to represent rainfall-driven soil water dynamics and especially preferential water flow and solute transport in the vadose zone. Our LAST-Model (**La**grangian **S**oil **W**ater and **S**olute **T**ransport) is based on a Lagrangian perspective on the movement of water particles (Zehe and Jackisch, 2016) carrying a solute mass through the subsurface which is separated into a soil matrix domain and a preferential flow domain. The preferential flow domain relies on observable field data like the average number of macropores of a given diameter, their hydraulic properties and their vertical length distribution. These data may either be derived from field observations or by inverse modelling using tracer data. Parametrization of the soil matrix domain requires soil hydraulic functions which determine the parameters of the water particle movement and particularly the distribution of flow velocities in different pores sizes. Infiltration into the matrix and the macropores depends on their respective moisture state and subsequently macropores are gradually filled. Macropores and matrix interact through diffusive mixing of water and solutes between the two flow domains which again depends on their water content and matric potential at the considered depths.

The LAST-Model is evaluated using tracer profiles and macropore data obtained at four different study sites in the Weiherbach catchment in south Germany and additionally compared against simulations using HYDRUS 1-D as benchmark model. While both models show an equal performance at two matrix flow dominated sites, simulations with LAST are in a better accordance with the fingerprints of preferential flow at the two other sites compared to HYDRUS 1-D. These findings generally corroborate the feasibility of the model concept and particularly the implemented representation of macropore flow and macropore-matrix exchange. We thus conclude that the LAST-Model approach provides a useful and alternative framework for a) simulating rainfall-driven soil water and solute dynamics and fingerprints of preferential flow as well as b) linking model approaches and field experiments. We also suggest that the Lagrangian perspective offers promising opportunities to quantify water ages and to evaluate travel and residence times of water and solutes by a simple age tagging of particles entering and leaving the model domain.

## 1 Introduction

Until now, the most commonly used hydrological models have been following an Eulerian perspective on the flow processes with a stationary observer balancing dynamic changes in a control volume. The alternative Lagrangian perspective with a mobile observer travelling along the trajectory of a solute particle through the system (Currie, 2002) has up to now only been used to simulate advective-dispersive transport of solutes (Delay und Bodin, 2001; Zehe et al., 2001; Berkowitz et al., 2006; Koutsoyiannis, 2010; Klaus and Zehe, 2011).

However, this particle tracking approach is mostly embedded in frameworks with Eulerian control volumes which still characterize the dynamics of the carrying fluid. Lagrangian descriptions of the fluid dynamics itself are only realized in a few models. But such a particle tracking framework may offer many advantages, especially at the coping of the challenges induced by preferential water flow and solute transport in structured heterogeneous soils.

Preferential flow has become a major issue in hydrological research since the benchmark papers of Beven and Germann (1982), Flury et al. (1994) and Uhlenbrook (2006). The term of preferential flow is used to summarize a variety of mechanisms leading to a rapid water movement in soils. The most prominent one is the flow through non-capillary macropores (Beven and Germann, 2013) where water and solutes travel in a largely unimpeded manner due to the absence of capillary forces and bypass the soil matrix (Jarvis, 2007). Macropores can be classified into e.g. earth worm burrows, channels from degraded plant roots or shrinkage cracks and all of them are not static in space nor time (e.g. Blouin et al., 2013; Nadezhdina et al., 2010; Palm et al., 2012; van Schaik et al., 2014; Schneider et al., 2018). Especially in rural areas and in combination with agrochemicals, macropore flow can be a dominant control on stream and groundwater pollution (e.g. Flury, 1996; Arias-Estévez et al., 2008). To understand such water and solute movements a combination of plot-scale experiments and computer models is commonly used (Zehe et al., 2001; Šimůnek and van Genuchten, 2008; Radcliffe and Šimůnek, 2010; Klaus et al., 2013). One of the most frequently used approaches to simulate water flow dynamics and solute transport is to use the Darcy-Richards and the advection-dispersion equation. Both equations fundamentally assume that solute transport is controlled by the interplay of advection and dispersion (Roth, 2006; Beven and Germann, 2013) and that the underlying soil water dynamics are dominated by capillary-driven diffusive flow. While the second assumption is well justified in homogeneous soils, it frequently fails in soils with macropores. Consequently, we separate at least two flow regimes in soils: the slow diffusive flow in the soil matrix and the rapid advective flow in the macropores. Partial mixing between these two flow regimes is non-trivial as it depends on the hydraulic properties of the macropore walls, the water content of the surrounding soil, actual flow velocities, hydrophobicity of organic coatings and much more. The inability of the Richards equation to simulate partial mixing between both flow regimes is well known and a variety of different models have been proposed to address this problem (Šimůnek et al., 2003; Beven and Germann, 2013). But most of them are still fundamentally based on the Darcy-Richards equation like the most prominent and well-established double-domain models like for instance the HYDRUS model of Šimůnek and van Genuchten (2008).

A promising alternative approach is provided by particle-based Lagrangian models for subsurface fluid dynamics. The first implementation of such a model for soil water dynamics is the SAMP model proposed by Ewen (1996a; b). SAMP represents soil water by a large number of particles travelling in an one-dimensional soil domain by means of a random walk which is based on soil physics and soil water characteristics. A more recent example is the two-dimensional MIP model of Davies et al. (2013) developed for hillslopes. Fluid particles travel according to a distribution function of flow velocities which needs to be estimated from tracer field experiments. Exchange of particles among the different pathways is conceptualized as random process following an exponential distribution of mixing times. Inspired by the SAMP model, Zehe and Jackisch (2016) conceptualized a Lagrangian model describing soil water flow by means of a non-linear space domain random walk. In line with Ewen (1996), they estimated the diffusivity and the gravity-driven drift term of the random walk based on the soil water retention curve ($\Psi(\theta)$) and the soil hydraulic conductivity curve ($k(\theta)$).

The particle-based Lagrangian model of Zehe and Jackisch (2016) initially assumed that all particles travel at the same diffusivity and velocity corresponding to the actual soil water content. But a comparison to a Richards solver revealed that this straightforward, naive random walk implementation highly overestimates infiltration and redistribution of water in the soil. The solution for this overestimation was to account for variable diffusive velocities. Now, particles in different pore sizes travel with various diffusivities, which are determined based on the shape of the soil hydraulic conductivity curve. This approach reflects the idea that the actual soil water content is the sum of volume fractions that are stored in different pore sizes and that the different pore sizes constitute flow paths which differ in both advective and diffusive velocities.

Recently, this model was advanced by Jackisch and Zehe (2018) with the implementation of a second dimension which contains spatially explicit macropores to simulate preferential flow. Within a macropore the velocity of each particle is described by interactions of driving and hindering forces. Driver is the potential energy of a particle while energy dissipation due to friction at the macropore walls dissipates kinetic energy and accordingly reduces particle velocities. With this approach, Jackisch and Zehe (2018) tried to make maximum use of observables for model parametrization. The assets of their echoRD model are a self-controlling macropore film flow and its ability to represent 2-D infiltration patterns. The drawback of echoRD is the huge computational expense. The simulation time is about 10 to 200 times longer than real-time.

The huge computational expense of the echoRD model is one main motivation for us to develop a Lagrangian approach which balances necessary complexity with greatest possible simplicity. The other motivation is the inability of all models mentioned above to simulate solute transport appropriately. This is essential for a rigorous comparison of the model with tracer data and to get closer to the simulation of reactive transport. Thus, the main objectives of this study are to:

1) Present a new routine for solute transport and diffusive mixing for well-mixed matrix flow conditions which is implemented into the model of Zehe and Jackisch (2016) and to test this approach against tracer data from plot-scale experiments carried out in the Weiherbach catchment (Zehe and Flühler, 2001b).

2) Extend the model by implementing a macropore domain accounting for preferential flow of water and solutes and related exchange with the matrix domain. In contrast to the echorRD model, we maintain the one-dimensional approach to keep the computational expense moderate.

The structure of our LAST-Model (**La**grangian Soil Water and **S**olute **T**ransport) is hence similar to a double-domain approach. The main asset is that flow and transport in both domains and their exchange are described by the same stochastic physics and that the macropore domain can be parametrized by observable macropore geometries. This fact may help to overcome the limiting assumptions of the Darcy-Richards and the advection-dispersion equation. The refined LAST-Model is tested by extensive sensitivity analyses to corroborate its physical validity. Further, it is also tested with four tracer infiltration experiments at different study sites in the Weiherbach catchment which are either dominated by well-mixed conditions (sites 23, 31) or preferential flow in macropores (sites Spechtacker, 33). For comparison, these four experiments are also simulated with HYDRUS 1-D.

**2 Concept and implementation of the LAST-Model**

**2.1 The Lagrangian model of Zehe and Jackisch (2016) in a nutshell**

The basis of our development is the Lagrangian model of Zehe and Jackisch (2016). It describes infiltration and water movement through a spatial explicit 1-D soil domain dependent on the effects of gravity and capillarity in combination with a spatial random walk concept. Water is represented by particles with constant mass and volume. The density of soil water particles in a grid element represents the actual soil water content $\theta(t)$ (m³ m⁻³) which reflects in turn the sum of the volume fractions of soil water that are stored in pores of strongly different sizes. Water particles travel at different velocities in these pores which are characterized by the shape of the hydraulic conductivity and water diffusivity curve. The curves are subdivided into $N_B$ bins, starting from the residual moisture $\theta_r$ stepwise to the actual moisture $\theta(t)$ using a step size of $\Delta\theta = \frac{(\theta(t)-\theta_r)}{N_B}$ (Figure 1). The particle displacement within the bins is described by Equation 1:

$$z_i(t+\Delta t) = z_i(t) - \left(\frac{k(\theta_r+i\cdot\Delta\theta)}{\theta(t)} + \frac{\partial D(\theta_r+i\cdot\Delta\theta)}{\partial z}\right)\cdot\Delta t + Z\sqrt{2\cdot D(\theta_r + i\cdot\Delta\theta)\cdot\Delta t} \qquad i=1,...,N \qquad (1)$$

Where $z$ is the vertical position (m), $k$ the hydraulic conductivity (m s⁻¹), $i$ the number of the current bin, $D$ the water diffusivity (m s⁻¹), i.e. the product of the hydraulic conductivity $k(\theta)$ and the slope of the soil water retention curve with the relation $\frac{\partial\psi}{\partial\theta}$ (m), $t$ the simulation time (s), $\Delta t$ the simulation time step and $Z$ a random, uniformly distributed number between [-1, 1]. The equation comprises two terms. The first one represents gravity-driven downward advection of each particle based on the hydraulic conductivity, the second one is the diffusive term driven by capillarity. According to Figure 1 and Equation 1, particles in coarse pores travel more rapidly at a higher hydraulic conductivity due to wet conditions. In smaller pores or during drier conditions the flow velocities are so small that the particles are in fact immobile. This binning of particle velocities and diffusivities also opens the opportunity to simulate rainfall infiltration under non-equilibrium conditions. To this end, infiltrating rainfall-event water is treated as second type of particles which initially travel at gravity-driven, rapid velocities in the largest pore fraction and experience a slow diffusive mixing with the pre-event water particles of the matrix during a characteristic mixing time. Test simulations revealed that the Lagrangian model can simulate water dynamics under equilibrium conditions in good accordance with a Darcy-Richards approach for three different soils. For a detailed description of the underlying model concept and the derivation of the equations see the study of Zehe and Jackisch (2016).

**2.2 Representation of solute transport in the LAST-Model**

In a first step we implement a routine for solute transport into the particle model by assigning a solute concentration $C$ (kg m⁻³) to each particle. This implies that a particle carries a solute mass which is equal to its concentration times its volume. Due to the particle movements through the matrix domain the dissolved mass experiences advective transport in every time step. Diffusive mixing among all particles is calculated after each displacement step by summing up the entire solute mass in a grid element and dividing it by the amount of all present water particles. The underlying assumption of perfect mixing among all particles in a grid element

requires a diffusive mixing time corresponding to the molecular diffusion coefficient, which is smaller than the time step $\Delta t$. The latter is ensured by a sufficiently fine subdivision of the soil matrix.

## 2.3 The macropore domain and representation of preferential flow

The second and main model extension is the implementation of a 1-D preferential flow domain considering the influence of macropores on water and solute dynamics. This requires four main steps:

1.  Design of a physically based structure of the preferential flow domain;
2.  Conceptualisation of the infiltration and partitioning of water into the two domains;
3.  Description of advective flow in the macropores;
4.  Conceptualisation of water and tracer exchange between the macropore and the matrix domain.

### 2.3.1 The preferential flow domain

We define a 1-D macropore or preferential flow domain (pfd) which is surrounded by a 1-D soil matrix domain with vertically distinct boundaries. In line with other Lagrangian models, we represent water as particles with constant mass and volume corresponding to their domain affiliation. As the vertical extent and volume of the pfd is much smaller than that of the matrix domain, the corresponding particles must be much smaller to ensure that an adequate number of particles travel within the pfd for a valid stochastic approach.

The pfd comprises a certain amount of macropores. Each macropore has the shape and structure of a straight circular cylinder with a predefined length $L_M$ (m) and diameter $dmac$ (m) containing spherically shaped particles (Figure 2a). Two of the most important geometrical properties of the pfd are the macropore diameter and the total number of macropores $nmac$ (-) as they scale exchange fluxes and determine several other characteristics like the total macropore volume. The macropore number, lengths and diameters can be directly measured in field experiments as described in section 3.2. From these observable parameters it is further possible to calculate additional pfd parameters like the total volume, stored water mass at saturation, the circumference and the flux rate. As we assume purely gravity-driven flow, the flux rate, the hydraulic conductivity of the pfd $k_{pfd}$ (m s$^{-1}$) and the advective velocity of a particle within the pfd $v$ (m s$^{-1}$) are assumed to be equal and can be calculated by the diameter as also described in section 3.2.

Our 1-D approach can of course not account for the lateral positions of the macropores but the pfd allows a depth distribution of macropores which is important for calculating the depth-dependent exchange with the matrix (section 2.3.4). To calculate the water content and tracer concentrations, the macropores of the pfd are vertically subdivided into grid elements of certain length $dz_{pfd}$ (m). Therefore, water contents and solute concentrations are regarded as averaged over these grid elements. Within a grid element of a macropore we assume cubic packing of a number of particles $N$ (cf. Figure 2a), each having a mass $m_P$ (kg) which is derived from the total water mass stored in a macropore when fully saturated. Based on this mass and the water density, the pfd particles are also geometrically defined by a diameter $D_P$ (m) and volume $V_P$ (m³).

In a cubic packing the particles are arranged in the way that the centres of the particles form the corners of a cube. The concept of cubic packing facilitates the calculation of the proportion of particles having contact to the lateral surface of a grid element. The rectangle in Figure 2a describes such a lateral surface of a grid element, with a height corresponding to the grid element length $dz_{pfd}$ and the circumference $C$ (m) as length, which can be obtained when a macropore grid element is cut open and its surface is laid-flat. The number of particles which

can be packed into this rectangle have then contact to the lateral surface of this grid element. The proportion of these contact particles on the total amount of particles roughly corresponds to the hydraulic radius scaling the wetted cross section with the wetted contact area in a macropore. Within the mixing process only the contact particles are able to infiltrate via the interface into the soil matrix.

5 **2.3.2 Infiltration and partitioning of water into the two domains**

As a 1-D approach does not allow an explicit, spatial distribution of the incoming precipitation water over the soil surface, we use an implicit, effective infiltration concept. The infiltration and distribution of water are controlled by the actual soil moisture and the flux densities driven by the hydraulic conductivity and the hydraulic potential gradient of the soil matrix as well as by friction and gravity within the macropores (Weiler, 10 2005; Nimmo, 2016; Jackisch and Zehe, 2018). For example, a soil matrix with a low hydraulic conductivity increases the proportion of water infiltrating the macropores as it preferentially uses pathways of low flow resistance.

In our model, we use a variable flux condition at the upper boundary of the soil domain dependent on the precipitation intensity. Incoming precipitation water accumulates in an initially empty fictive surface storage 15 from which infiltrating water masses and related particle numbers are calculated. To this end, we distinguish several cases. In Case 1, the top soil grid elements of the soil matrix and the pfd are initially unsaturated and the infiltration capacity of the soil matrix is smaller than the incoming precipitation flux density. Water infiltrates the soil matrix and the excess water is redistributed to the pfd and infiltrates it with a macropore-specific infiltration capacity. Case 2 applies when the top matrix grid element is saturated and water exclusively infiltrates the pfd 20 until all macropores are also saturated. Case 3 occurs when both the top matrix layer and the pfd are saturated leading to an accumulation of precipitation water in the surface storage. As soon as the water contents in the first soil matrix grid element and in the pfd are subsequently decreasing due to downward water flow or drainage of the macropores, again infiltration occurs according to Case 1. The incoming precipitation mass ($m_{rain}$) and the infiltrating water masses into the matrix ($m_{matrix}$) and the pfd ($m_{pfd}$) are calculated with Equations 2-4. Please note 25 that these equations present infiltrating masses and not fluxes because the model generally works with discrete particles and their masses.

$$m_{rain} = q_{rain} \cdot \rho_w \cdot \Delta t \cdot A \qquad (2)$$

$$m_{matrix} = \left(\frac{k\_m_1 + k_s}{2}\right) \cdot \left(\frac{\psi_1 - \psi_2}{dz} + 1\right) \cdot A \cdot \rho_w \cdot \Delta t \qquad (3)$$

30 $$m_{pfd} = k_{pfd} \cdot \pi \cdot \left(\frac{dmac}{2}\right)^2 \cdot \rho_w \cdot \Delta t \cdot nmac \qquad (4)$$

Where $q_{rain}$ (m s$^{-1}$) is the precipitation flux density, respectively the intensity, $k\_m_1$ (m s$^{-1}$) the actual hydraulic conductivity of the first grid element of the matrix, $k_s$ (m s$^{-1}$) the saturated hydraulic conductivity of the matrix and $\psi_1 - \psi_2$ (m) the matric potential difference between the surface and the first grid element right beneath the 35 surface, $dz$ (m) the grid element length in the matrix domain (0.1 m), $k_{pfd}$ (m s$^{-1}$) the saturated hydraulic conductivity of a macropore (cf. section 3.2), $dmac$ (m) the diameter of a macropore and $nmac$ (-) the total

number of macropores within the pfd, $\rho_w$ (kg m$^{-3}$) the water density, $\Delta t$ (s) the simulation time step and $A$ (m²) the plot area.

According to Equation 3, the infiltration rate into the matrix is based on Darcy's law and thus we are generally able to account for an extra pressure due to a ponded surface, e.g. in Case 3. But in our simulation cases, ponding heights are small and have only marginal effect. After the precipitation water has infiltrated into the two domains the masses are converted to particles which are initially stored in the first grid elements of the matrix and pfd. They are now ready for the displacement process.

### 2.3.3 Advective flow in the macropores

In the pfd, we assume a steady state balance between gravity and dissipative energy loss at the macropore walls. This implies purely advective flow characterised by a flow velocity $v$ which can either be inferred from tracer or infiltration experiments on macroporous soils as described by Shipitalo and Butt (1999), Weiler (2001) and Zehe and Blöschl (2004). The particle displacement in our pfd is described by Equation 5:

$$\Delta z = v \cdot \Delta t \tag{5}$$

As all particles in the pfd travel at the same velocity, their displacement depends on the time step. Generally, our model can work with variable time stepping as Lagrangian approaches are not subject to time step restrictions or numerical stability criteria. Here, we select the time step such that the particle displacement per time step equals the maximum depth of the pfd and subsequently excess particles are shifted upwards to the deepest unsaturated grid element. In this way, we gradually fill the macropores from the bottom to the top comparable to the filling of a bottle with water. This simple volume filling method was applied before in other models, e.g. in the SWAP model of van Dam et al. (2008) or in the study of Beven and Clarke (1986). Figure 2b shows an example for the macropore filling concept: in each of the three points in time (t1-t3) new particles, shown by the different colours, infiltrate the macropore and subsequently they are displaced with $\Delta z$ to the bottom of the macropore, initially saturating the deepest grid element (t1). In the following points in time t2 and t3 the new particles do not fit into the respective saturated grid elements anymore and are then shifted to the next deepest unsaturated grid element. In line with the matrix, particle densities are calculated in each grid element to obtain the actual soil water content and tracer concentrations of the pfd.

### 2.3.4 Water and tracer exchange between the macropore and the matrix domain

Commonly, macropore-matrix interactions are challenging to observe within field experiments. One approach is to evaluate the isotopic composition of water in the two domains (Klaus et al., 2013). In theory it is often assumed that the interactions and water dynamics at the interface between macropores and the matrix are mainly controlled by the matric head gradients and the hydraulic conductivity of both domains which depend on an exchange length and the respective flow velocities (Beven and Germann, 1981; Gerke, 2006).

Our model approach is also based on these assumptions as illustrated in Figure 2c. We restrict exchange to the saturated parts of the pfd assuming downward particle transport as being much larger than the lateral exchange and we neglect diffusive exchange between solutes in the matrix and the pfd. We are aware that these simplifications might constrain the generality of our model. For instance, we also neglect the effect of a reverse

diffusion from the matrix into the macropores. This effect can influence water and solute dynamics when the propagation of a pressure wave pushes matrix water into empty macropores, mainly in deeper saturated matrix areas (Beven and Germann, 2013). We rely on those simplifications a) to keep the model simple and efficient and b) because the focus of our model is on unsaturated soil domains and during rainfall-driven conditions the

macropores are most of the time largely filled due to their small storage volume.

The distribution of different macropore depths and the definition of distribution factors can be derived from datasets containing information on macropore networks observed in field experiments as described in section 3.2. Based on these datasets, the current version of our model divides the total amount of macropores $nmac$ in the pfd into three depths. To this end, the total number is multiplied with a distribution factor $f$ for big ($f_{big}$),

medium ($f_{med}$) and small ($f_{sml}$) macropores (cf. Figure 2c).

The saturated grid elements (blue filled) of the largest macropores are coupled to the respective grid elements of the medium and small macropores. In this example, the red respectively the black framed grid elements of the three macropore sizes are coupled due to their saturation state and depth order. This coupling ensures a simultaneous diffusive water flow out of the respective grid elements of all three macropore depths. The mixing

fluxes ($q_{mix}$ (m s$^{-1}$)) in the actual grid elements are calculated by Equation 6:

$$q_{mix} = \frac{2 \cdot k_s \cdot k\_m_i}{(k_s + k\_m_i)} \cdot \frac{\psi_i}{dmac} \cdot C \cdot dz_{pfd} \tag{6}$$

Thus, diffusive mixing fluxes are calculated with the harmonic mean of the saturated hydraulic conductivity of

the matrix $k_s$ (m s$^{-1}$) and the current hydraulic conductivity of the respective matrix grid element $k\_m_i$ (m s$^{-1}$), multiplied with the relation of the matric potential $\psi_i$ (m) of the actual matrix grid element and the macropore diameter $dmac$ (m) as exchange length and the circumference $C$ (m) of the macropore grid element. We use the harmonic mean here because we assume a row configuration at the calculation of the lateral diffusive mixing fluxes between macropore and matrix as there is a vertical interface between the two domains.

The mixing masses are again converted into particle numbers with the two different particle masses. Due to the higher masses of the matrix particles a much lower amount of particles is entering the matrix. This has to be taken into account by choosing an adequate number of total particles present in the matrix, i.e. at least one million at moderate saturated hydraulic conductivities. In addition, it is ensured that the number of particles leaving a grid element of the pfd is lower than the maximum possible number of particles having contact to the

lateral surface (cf. section 2.3.1) dependent on its current soil water content. Please note that up to now our model works with a no-flow condition at the lower boundary of the pfd but the model structure is generally capable to add an additional diffusive drainage with particles leaving the macropores at their lower boundary.

**3 Model benchmarking**

**3.1 Evaluation of the solute transport and linear mixing approach during well-mixed matrix flow**

Basis of the first evaluation of our solute transport and linear mixing approach are data from tracer experiments conducted by Zehe and Flühler (2001b) in the Weiherbach catchment to investigate mechanisms controlling flow patterns and solute transport. The Weiherbach valley is located in the southwest of Germany and has a total extent of 6.3 km². The basic geological formations comprise Triassic Muschelkalk marl and Keuper sandstone

covered by Pleistocene Loess layers with a thickness of up to 15 m. The hillslopes exhibit a typical Loess catena with erosion derived Colluvic Regosols at lower slopes and Calcaric Regosols or Luvisols at the top and mid slopes. Land use is dominated by agriculture. For further details on the Weiherbach catchment please see the work of Plate and Zehe (2008).

In this catchment, a series of irrigation experiments with bromide as tracer were performed at ten sites. At each site, a plot area of 1.4 m x 1.4 m was defined and the initial soil water content and the soil hydraulic functions were measured. The plot area was then irrigated by a block rainfall of approx. 10 mm h$^{-1}$ with a tracer solution containing 0.165 kg m$^{-3}$ bromide. After one day, soil profiles were excavated and soil samples were collected in a 0.1 m x 0.1 m grid down to a depth of 1 m and their corresponding bromide concentrations measured.

Thus, every 10 cm soil depth interval, ten samples were taken and for the comparison with our 1-D simulation results, the bromide concentrations were averaged over each sample depth. Note that the corresponding observations provide the tracer concentration per dry mass of the soil $C_{dry}$ while the LAST-Model simulates concentrations in the water phase $C_w$. We thus compare simulated and observed tracer masses in the respective depths. More details on the tracer experiments can be taken from Zehe and Flühler (2001a; b). For the evaluation

of our solute transport and linear mixing approach, we select the two sites 23 and 31 where flow patterns reveal a dominance of well-mixed matrix flow without any considerable influence of macropores. Thus, we use the LAST-Model without an active pfd for the simulations at the study sites 23 and 31.

The soil at the two sites can be classified as Calcaric Regosol (IUSS Working Group WRB, 2014). In line with the experiments, our model uses a spatial soil matrix discretization of 0.1 m and the soils initially contain in total

20 1 million water particles but with no tracer masses. Initial soil water contents and all further experimental and model parameters as well as the soil properties at these sites are listed in Table 1.

**3.2 Parametrization and evaluation of the preferential flow domain**

In a next step, our pfd model extension is again evaluated with the help of the results of two additional field tracer experiments of Zehe and Flühler (2001b). This time, we select the study sites Spechtacker and 33 which

show numerous worm burrows inducing preferential flow. The sites are also located in the Weiherbach catchment and the sprinkling experiments were equally conducted with the application of a block rainfall containing bromide on a soil plot. The soils can be classified as Colluvic Regosol (IUSS Working Group WRB, 2014).

Additionally, the patterns of the worm burrows were extensively examined at these study sites. Horizontal layers

in different depths of the vertical soil profiles were excavated (cf. introduction of van Schaik et al., 2014) and in each layer the amount of present macropores counted as well as the diameters and depths measured. These detailed measurements provided an extensive dataset of the macropore network at the study sites Spechtacker and 33. Based on this dataset, we can obtain those data we need for the derivation of a mean macropore diameter, macropore depth distribution and distribution factors. We focus on a mean macropore diameter of 5

35 mm at the site Spechtacker because worm burrows with a diameter range of roughly 4 - 6 mm are dominant here and at site 33 we select a mean diameter of 6 mm. Figure 3 shows the mean number of macropores with these diameters in each depth at both sites. Based on this distribution, we can identify and select three considerable macropore depths at the site Spechtacker (0.5 m, 0.8 m and 1.0 m) and two macropore depths at site 33 (0.6 m and 1.0 m) (cf. Table 1). In these depths, we count circa 11, 3 and 2 macropores *(nmac* = 16) at the site

Spechtacker as well as 30 and 16 macropores (*nmac* = 46) at site 33, respectively. With these distributions we are able to calibrate our distribution factors *f* in the way that a multiplication of the total number of macropores with these factors results in the correct number of macropores in the respective depths. The obtained distribution factors are listed in Table 1.

Moreover, Zehe and Flühler (2001b) measured saturated water flow through a set of undisturbed soil samples containing macropores of different radii at the study site Spechtacker with the assumption that flow through these macropores dominated. In line with the law of Hagen-Poiseuille, they found a strong proportionality of the flux through the macropores to the square of the macropore radius while frictional losses were 500 to 1000 times larger. This dependence of the flux rate on the macropore radius can be described by the linear regression shown

in Figure 4. Based on this linear regression, the hydraulic conductivity of the macropores $k_{pfd}$ was calculated as a function of the macropore radius $\frac{dmac}{2}$ (termed $r_M$ in Zehe and Flühler, 2001b) as we assume the hydraulic conductivity $k_{pfd}$ is equal to the flux rate $q_M$ of the macropore (Equation 7).

$$k_{pfd} = 2884.2 \cdot \left(\frac{dmac}{2}\right)^2 \tag{7}$$

For more details on the two study sites and their macropore network, see also the studies of Ackermann (1998) and Zehe (1999). Here, we select a spatial pfd discretization of 0.05 m and assume that macropores initially contain no particles and hence also no water or tracer masses. The total possible number of particles which can be stored in the pfd is 10,000 particles. All further experimental and simulation parameters, soil properties as

well as information about the macropore network at the sites Spechtacker and 33 are listed in Table 1.

**3.3 Simulations with HYDRUS 1-D**

The simulations with HYDRUS 1-D are performed with the same soil properties, model setups and initial conditions introduced in the sections 3.1 and 3.2 as well as shown in Table 1. The simulations of the well-mixed sites 23 and 31 are performed with a van Genuchten - Mualem single porosity model for water flow and an

equilibrium model for solute transport. For the simulations at the preferential flow sites Spechtacker and 33 we use dual-porosity models for water flow ("Durner, dual van Genuchten – Mualem") and solute transport ("Mobile - Immobile Water"). This means HYDRUS assumes two differently mobile domains to account for preferential flow. The theory of that approach describes preferential flow in the way that the effective flow space is decreased due to the immobile fraction and thus the same volume flux is forced to flow through this decreased

flow space resulting in higher pore water velocities and consequently also in a deeper percolation of water and solutes (Šimůnek and van Genuchten, 2008). For the parametrization of these two domains we select an immobile soil water content *ThImob.* of 0.2 m³ m⁻³. We hence assume that about 80 – 90 % of the total soil water amounts at the two sites are stored in the matrix and are therefore in fact immobile compared to the remaining 10 – 20 %, which are assumed to flow through macropores. Zehe and Jackisch (2016) elaborated this rate of an

immobile and mobile fraction in the fine-grained soils of the Weiherbach catchment. For all simulations we choose an atmospheric condition with a surface layer and variable infiltration fluxes at the upper boundary as well as a free drainage condition at the lower boundary.

### 3.4 Sensitivity analyses of selected parameters

The sensitivity analyses of the model with the pfd-extension are conducted by varying several parameters describing the soil matrix and the pfd in a realistic, evenly spaced value range. To this end, the saturated hydraulic conductivity of the matrix $k_s$, the diameter *dmac* and the number *nmac* of the macropores are the selected parameters which are deemed to be most sensitive and crucial for the model behaviour and the simulation results. The probably most sensitive parameter is $k_s$ as it controls the infiltration capacities of both domains, the displacement within the soil matrix as well as the diffusive mixing fluxes. Beside the saturated hydraulic conductivity of the matrix, we also assume that the total number and diameter of the macropores are probably of great importance for the model results because they are crucial for the development of the new pfd (cf. section 2.3.1). Moreover, based on the derived three depths and distribution factors at the site Spechtacker (cf. section 3.2) we arbitrarily select different configurations of the macropore depth distribution and the distribution factors to evaluate the behaviour of the model related to various numbers of macropores in different depths. The depth distribution of macropores thereby comprises a deep (Configuration 1), medium (Configuration 2) and shallow (Configuration 3) distribution. At the distribution factors there are four different configurations. A realistic distribution comprising more small than big macropores is represented by Configuration A and D, a homogeneous distribution is shown by Configuration B and a rather uncommon distribution with more big than small macropores is illustrated by Configuration C. All parameter ranges and the detailed configurations of the sensitivity analyses are listed in Table 2.

All model runs of the sensitivity analyses are performed at the site Spechtacker using 22 mm of rainfall in 140 minutes with subsequent drainage duration of one day. Additional parameters like soil properties, antecedent moisture and concentration states, bromide concentration of precipitation water remain constant (cf. Table 1).

## 4 Results

### 4.1 Simulation of solute transport under well-mixed conditions

The well-mixed sites 23 and 31 show a high similarity due to their spatial proximity (Figure 5a, b). The shape and courses of the simulated tracer mass profiles coincide well with the observed ones over the entire soil domain with RMSE values of 0.23 g and 0.28 g, respectively. The observed values are within the uncertainty range, represented by the rose shaded areas. This area reflects the uncertainty arising from a variation of $k_s$ values of the soil matrix in the observed range of $10^{-7}$ - $10^{-6}$ m s$^{-1}$ at site 23 and $10^{-6}$ - $10^{-5}$ m s$^{-1}$ at site 31.

Note that in the experiments the tracer mass was not directly measured at the soil surface but the observations represent averages across 10 cm depth increments, starting in a depth of 5 cm. A comparison of the simulated masses close to the surface is thus not meaningful. This difference between simulated and observed profiles near to the surface suggests that the coarse resolution of the sampling grid is a likely reason for the relatively low recovery rates of 77 % and 76 % at the two sites (cf. Table 1). Overall, we conclude that manipulating $k_s$ within the observed uncertainty leads to an unbiased simulation ensemble compared to the observed tracer data at matrix flow dominated sites.

## 4.2 Evaluation of the preferential flow domain

Our model with the new preferential flow domain is tested against two tracer experiments on macroporous soils at the sites Spechtacker and 33. At the site Spechtacker, the simulated and observed tracer mass distributions are generally in good accordance (Figure 6a) with a RMSE of 0.3 g and again the values are within the uncertainty range. In this case, the rose area shows the standard deviation of measured macropore numbers ($\pm$ 4) and diameters ($\pm$ 1 mm) from the mean values (cf. Table 1) at the site Spechtacker. Especially in deeper soil regions from 0.35 m to 1 m, the shape and the magnitude of values correspond well. In the upper soil parts from 0.05 m to 0.15 m the model slightly overestimates the tracer masses. Between 0.15 m and 0.35 m soil depth both profiles exhibit the greatest differences and even contrary courses.

In general, the simulated mass profile at site 33 corroborates the results of the site Spechtacker (Figure 6b). The simulated and observed tracer masses are also in a good accordance with a RMSE value of 0.15 g. In contrast to the site Spechtacker, varying the macropore numbers and diameters within the standard deviation ($\pm$ 4; $\pm$ 1 mm) has just slight effects on the mass profile at this site. However, especially in deeper soil regions from 0.6 m to 1 m the values correspond well, while the greatest differences occur between 0.25 m and 0.45 m as the simulated mass profile is not able to completely depict the observed hump in this area.

## 4.3 Comparison with HYDRUS 1-D

The mass profiles at the well-mixed sites 23 and 31 simulated with HYDRUS 1-D show similar patterns and are in accordance with the observed profiles with RMSE values of 0.1 g at site 23 and 0.37 g at site 31 (Figures 5c, d). Especially at site 23 the simulated mass profile is centred within the uncertainty range of the measured $k_s$ values (rose shaded area, cf. section 4.1). At site 31, HYDRUS 1-D slightly overestimates the tracer masses over the entire soil domain but also here the shape of the profiles coincide well. In contrast, at the two preferential flow sites Spechtacker and 33 the mass profiles simulated with HYDRUS 1-D and the dual-porosity approach (rose profile) are not in a good accordance with the observed profiles with RMSE values of 0.46 g and 0.53 g, respectively (Figures 6c, d). In the first 40 cm there is an overestimation of the simulated tracer masses, while in the deeper soil regions HYDRUS 1-D is not able to reproduce well the tail of the mass profiles with their heterogeneous courses. A comparison with the results of HYDRUS with an equilibrium model (red profile) reveals that the dual-porosity approach is generally able to predict a deeper percolation of solutes through the mobile domain.

## 4.4 Sensitivity analyses

### 4.4.1 Sensitivity to saturated hydraulic conductivity $k_s$

The concentration profile range of the matrix reveals a strong sensitivity of the simulated profiles to $k_s$ when we neglect macropores (Figure 7a). Especially in the upper soil part, the differences arising from low and high $k_s$ values are clearly detectable. Lower values imply that the soil matrix has a smaller infiltration capacity and therefore less water is infiltrating the matrix. Consequently, without macropores solutes do not penetrate into depths greater than 0.2 m. The presence of macropores significantly alters the sensitivity of the concentration and soil moisture profiles (Figures 7b, c). Again, the profile shapes clearly depend on the $k_s$ values but now water and solutes reach greater depths of down to 0.8 m by flowing through the macropores. At low $k_s$ values (red

curve) the reduced matrix infiltration capacity leads to an increased infiltration of water and solute into the macropores. Subsequently, the solutes bypass the matrix until they diffusively mix into the matrix at greater depths.

In contrast, at high $k_s$ values the matrix infiltration capacity is increased. This leads in turn to a reduced infiltration into the macropores and instead the majority of water and solute masses infiltrates the matrix and remains in the top soil. This effect is reflected by the blue curves in Figure 7 with higher solute concentrations near the soil surface and decreased concentrations at greater depths in comparison to low $k_s$ values.

Finally, the yellow curves in Figure 8 show the proportion of solutes within the matrix which originates from the macropores. In general, at all $k_s$ values and depths below 0.2 m the entire solute amount within the matrix travelled through the macropores. Differences are restricted to the upper soil part. Here the largest proportion of solutes has directly infiltrated the matrix without having been in the macropores before. The pfd proportion decreases from low to high $k_s$ values confirming again the important influence of the $k_s$ values on the infiltration capacities and the distribution of water and solutes.

### 4.4.2 Sensitivity to macropore number *nmac* and diameter *dmac*

The model results sensitively respond to a variation of macropore diameters. In the upper soil part, the solute concentrations and moisture are slightly higher, when macropores are small (Figures 9a, b). In this case, the macropores collect only smaller amounts of water and solutes and the majority has directly infiltrated the soil matrix. Wider macropores transport larger amounts of water and solutes to greater depths where they diffusively mix into the subsoil matrix. This deep redistribution is reflected by the characteristic profile shapes and the higher concentration and moisture values in the deep soil.

Furthermore, the influence of different macropore numbers on the concentration and moisture profiles is marginal (Figures 9c, d). This implies that the model does not respond to every geometrical parameter equally sensitively. The macropore number scales less than the diameter at the calculation of the further macropore measures. However, this could change when working with higher precipitation intensities.

Simulations with different macropore depth configurations again reveal a clear sensitivity of the model (Figures 10a, b). A steady decrease of the deep redistribution of the concentration and moisture values from the deep (Configuration 1) to the shallow depth configuration (Configuration 3) is obvious. Shallow macropores distribute the total amount of water and solutes mainly in the upper soil part, while deep macropores relocate this distribution to greater depths of down to 1 m. The results of the distribution factor configurations again corroborate the previous findings (Figures 10c, d). Configuration B produces a homogeneous solute concentration profile from 0.2 m to the total depth. Both more realistic Configurations A and D comprise more small than big macropores. This increased number of small macropores ensures higher water and solute amounts in the first 0.5 m of the soil matrix due to an enhanced mixing in this area. Finally, the rather uncommon Configuration C with more big than small macropores shows converse results. Solute concentrations and moisture contents are strongly increased at great depths from 0.7 m to 1 m because of increased diffusive mixing fluxes in these parts.

**5 Discussion and Conclusions**

We extend the Lagrangian model of Zehe and Jackisch (2016) with routines to consider transport and linear mixing of solutes within the soil matrix as well as preferential flow through macropores and related interactions with the soil matrix. The evaluation of the model with data of tracer field experiments, the comparison with results of HYDRUS 1-D and the sensitivity analyses reveal the feasibility and physical validity of the model structure as well as the robustness of the solute transport and linear mixing approach. The LAST-Model provides a promising framework to improve the linkage between field experiments and computer models to reduce working effort, and to improve the understanding of preferential flow processes.

**5.1 New routine for solute transport and diffusive mixing**

The initially performed simulations of the bromide mass profiles at the two well-mixed sites 23 and 31 support the validity of the straightforward assumptions of the underlying solute transport routine with its perfect mixing approach (Figures 5a, b). In the presented version, our mixing routine works with a short mixing time to ensure an instantaneous mixing between event and pre-event particles to account for the well-mixed conditions at the selected sites. However, the model allows to select longer mixing times or even a distribution of various mixing times to consider imperfect mixing among different flow paths.

The simulation results at the well-mixed sites 23 and 31 are confirmed by the commonly approved HYDRUS 1-D model. The simulated tracer mass profiles and RMSE values of both models are in a good accordance at these sites (Figure 5). The capability of predicting the solute dynamics is hence a big asset of our approach and it is a solid base to realize the second model extension with the implementation of the preferential flow domain.

**5.2 Model extension to account for preferential flow in macropores**

The results of the evaluation of the pfd-extension show that our model is furthermore capable to simulate tracer experiments on macroporous soils and to depict well their observed 1-D tracer mass profiles with the typical fingerprint of preferential flow (Figure 6a, b). Especially the tracer masses in the subsoil match well between simulated and observed data. This corroborates our assumptions concerning the macropore structure and the approach to describe macropore-matrix exchange which proved to be feasible to predict solute distribution patterns due to preferential flow and related long transport lengths. In this context, we stress that the approach to simulate macropore-matrix exchange (cf. Figure 2c) does not rely on an extra leakage parameter but follows the theory of deriving an effective diffusive exchange between the domains (cf. Equation 6).

In contrast, the HYDRUS 1-D model performs clearly inferior and does not match the fingerprints of preferential flow in the mass profiles at the sites Spechtacker and 33 (Figures 6c, d). Especially the penetration of bromide through macropores into greater depths is ignored by HYDRUS 1-D, although we selected dual-porosity models for both water flow and solute transport (cf. section 3.3). The better performance of our LAST-Model at the two preferential flow sites compared to HYDRUS is further reinforced by the RMSE values which are significantly different. The results imply that, when working with a dual-porosity approach, HYDRUS and the underlying theory of two differently mobile domains is indeed able to depict a generally deeper penetration of solutes but it is not sufficient to exactly simulate the heterogeneous course and shape of the observed tracer mass profiles in preferential flow dominated soil domains.

The results of our LAST-Model mainly deviate from the observations in the upper soil parts. However, these deviations are within the uncertainty ranges revealed by the sensitivity analyses (Figures 7, 9). Further, the model reveals difficulties at the simulation of bromide masses between 0.15 m and 0.35 m soil depth at the site Spechtacker (Figure 6a). Possible reasons could be the influence of a) lateral endogeic worm burrows which are completely unknown and not represented in the model and b) a nearby plow horizon. Both reasons result in a disturbance of the soil structure leading to an increased uncertainty of soil properties in this region.

At site 33, our model is not able to sufficiently reproduce the hump of the observed mass profile between 0.25 m and 0.45 m soil depth (Figure 6b). A possible explanation for this issue could be the fact that the tracer experiment and the examination of the macropore network were performed on different dates. It is likely that uncertainties arise from this temporal discrepancy with a mismatch between observed macropore geometries and recovered tracer pattern due to natural soil processes as well as anthropogenic soil cultivation during this time lapse. Another possible explanation could be the fact that up to now the exchange is only simulated for saturated parts of the pfd (cf. section 2.3.4) and hence the transport of solute masses from the pfd into the matrix is delayed. A test of this idea requires a refinement of the model in future research. Moreover, varying macropore numbers and diameters in the range of the standard deviation reveals just slight effects on the simulated mass profile at site 33 and is thus less sensitive compared to the results at the site Spechtacker. The reason for this phenomenon is probably the higher total number of macropores ($nmac = 46$) and thus a larger macropore volume at site 33. In relation to this larger volume, the variation of macropore numbers and diameters in the quite narrow range of the standard deviation ($\pm 4$, $\pm 1$ mm) has only a minor influence on the total water and tracer masses transported through the macropore network and thus on the resulting mass profile at site 33.

Note that the conversion of solute masses into an integer number of particles results in small errors, leading to a small amount of solutes not entering the system and remaining in the fictive surface storage. To mitigate this model effect, a high number of total particles present in the matrix is necessary, at least one million. Beside many displacement steps of each particle, the total number of particles is important to render the random walk approach statistically valid (Uffink, 1990), although too high particle numbers will decrease the computational efficiency. Thus, we conclude that our extension of the Lagrangian particle model of Zehe and Jackisch (2016) is a promising tool for a straightforward 1-D estimation of non-uniform solute and water dynamics in macroporous soils. However, before the suitability of our model approach to simulate preferential flow of non-interacting tracers is generalized, further field experiments on a variety of differently structured soils are necessary. In the presented model version, we assume that a macropore distribution with maximal three different depths is a sufficient approximation of the observed macropore networks at the study sites Spechtacker and 33 (cf. section 3.2, Figure 3). Nevertheless, as a variable macropore depth distribution might be observed at other sites, the implementation of the macropore depth distribution must be kept flexible for other soils in future model parametrizations. Beside the parametrization with experimental data, it is also possible to setup our model by using pedotransfer functions for the soil hydraulic properties and to vary the parameters of the pfd by inverse modelling, which needs prior knowledge of the depth of typical macropore systems (e.g. worm burrow networks) and literature data to parametrize macropore flow velocities. This method would reduce time and the amount of work but it could result in equifinality as shown by Klaus and Zehe (2010) or Wienhöfer and Zehe (2014).

Some of our assumptions like the macropore geometry, the simple volume filling or the depth distribution of macropores were applied in a similar way in dual-porosity models before (Beven and Germann, 1981; Workman and Skaggs, 1990; van Dam et al., 2008) and even few previous studies also worked with physically and geometrically separated domains (e.g. Russian et al., 2014). Thus, our model extension can be seen as an advancement of double-domain approaches by assuming simple volume filling for macropore flow and particle tracking for matrix flow instead of relying on the Darcy-Richards equation. With these results, our model is one of the first which proves that simulations based on a Lagrangian perspective on both solute transport and dynamics of the carrying fluid itself are possible and well applicable. Also, the vertically distributed exchange between both domains seems feasible and does not rely on extra parameters like a leakage coefficient, e.g. used in dual-models (Gerke, 2006). The concept of cubic particle packing within the macropores (cf. Figure 2a, section 2.3.1) is strongly motivated by the hydraulic radius and can thus be transferred to flow in further kinds of macropore geometries, including flow between two parallel walls like it occurs in soil cracks or corner flow in rills (Germann, 2018).

Another remarkable result is the high model sensitivity towards the saturated hydraulic conductivity $k_s$ of the soil matrix (Figures 7, 8). Especially its direct influence on the infiltration process is crucial. As $k_s$ determines the initialisation, infiltration fluxes and the distribution of incoming precipitation masses to the two domains, it has a direct impact on the deep displacement of water and solutes. Therewith, our findings highlight the importance of infiltration processes on macroporous soils and the challenge to implement them properly into models which have also been stressed by other studies (Beven and Germann, 1982; Weiler, 2005; Nimmo, 2016).

Our model shows further a remarkable sensitivity to the presence of a population of macropores while differences in macropore properties comparatively have little impact. Generally, wider macropores collect and transport more water and solutes to greater depths than small ones (Figures 9a, b). In contrast, high numbers of macropores do not necessarily result in a greater and deeper percolation of solutes (Figures 9c, d). Jackisch and Zehe (2018) also reported this aspect and explain it with the distribution of the irrigation supply to all macropores and this supply can drop below the diffusive mixing fluxes from the macropores into the matrix. However, this implies that the number of macropores becomes more sensitive at much larger irrigation rates.

Where and to which extent water and solutes are diffusively mixed from the macropores into the matrix clearly depends on the depth distribution of the macropores and the distribution of the mixing masses among the various depths (Table 2, Figure 10). This concept of the distribution of macropore depths and mixing masses is important to meet the natural condition of a high spatial heterogeneity of the macropore network. Generally, the results of our sensitivity analyses are in line with the findings of Loritz et al. (2017) as they reveal a significant impact of the implementation of macropore flow on the model behaviour and its complexity.

Please note that we are aware of the fact that some results of the sensitivity analyses are straightforward and expectable. Nevertheless, we think that their presentation is necessary to allow the reader to check if our Lagrangian approach with the macropore domain reproduces these results as the model concept is new. To this end, please also see further sensitivity analyses in the appendix.

We overall conclude that the modified 1-D structure of our model is robust and provides a high computational efficiency with short simulation times, which is a big advantage of our model. In line with the underlying

Lagrangian model of Zehe and Jackisch (2016), we also used the programming language MATLAB to develop the two model extensions. The model simulation at the site Spechtacker with the selected parametrization (cf. Table 1) only runs for about five minutes, even on a personal computer with moderate computing power. Without an active pfd, like it is the case for the simulations at the study sites 23 and 31, the model runs even

faster. If performing these simulations on a high performance computer or workstation, one could probably also run several model simulations in parallel within minutes.

Moreover, the efficiency allows for the implementation of further routines with yet still appropriate simulation times. In this way, the model could prospectively consider retardation and adsorption effects as well as first-order reactions during the transport of non-conservative substances like pesticides. Until now, the solute

movement of conservative tracers like bromide is only determined by the water flow without any consideration of molecular diffusion or particle interactions, although some evidence suggests a non-conservative behaviour of bromide tracers under certain conditions (e.g. Whitmer et al., 2000; Dusek et al., 2015). In our case, we believe that the event scale and the short simulation times allow for the assumption of a conservative behaviour of bromide.

Moreover, the model can be extended to 2-D for simulations on hillslope or even catchment scales. In this regard, our model also offers the promising opportunity to quantify water ages and to evaluate travel and residence times of water and solutes by a simple age tagging of particles. This can shed light on the chemical composition and generation of runoff fluxes as well as on the "Inverse Storage Effect". This effect describes a greater discharge fraction of recent event water at a high catchment water storage than at low storage

(Hrachowitz et al., 2013; Harman, 2015; Klaus et al., 2015; van der Velde et al., 2015; Sprenger et al., 2018).

*Data availability.* The LAST-Model, reference data and the presented test experiments are available in a GitHub repository: https://github.com/KIT-HYD/last-model.

*Competing interests.* The authors declare that they have no conflict of interest.

*Acknowledgements.* This research contributes to the Catchments As Organized Systems (CAOS) research group (FOR 1598) funded by the German Science Foundation (DFG).

The article processing charges for this open-access publication were covered by a Research Centre of the Helmholtz Association.

**Appendix: Further sensitivity analyses with time series**

We performed additional sensitivity analyses to determine the effect of different $k_s$ values and macropore diameters on the temporal development of the solute concentration profile. We moved the results of these time series to the appendix as they generally provide no new insights but confirm the findings presented in the results

section.

Figure A1 generally confirms the findings of the sensitivity analyses with different $k_s$ values (cf. section 4.4.1). The four temporal snapshots show the development of the concentration profiles at low ($1 \cdot 10^{-6}$ m s$^{-1}$), medium ($2.5 \cdot 10^{-6}$ m s$^{-1}$) and high ($1 \cdot 10^{-5}$ m s$^{-1}$) $k_s$ values throughout the simulation time with a) + b) during the rainfall

event and c) + d) shortly after it and after one day, respectively. It is obvious how rapidly solute concentrations increase, especially in the upper soil part at high $k_s$ values. Shortly after the rainfall event almost the entire water and solute masses have infiltrated the matrix due to the higher infiltration capacity. At low $k_s$ values, water and solutes notably need more time to infiltrate completely. The differences of the centres of mass and the deeper shift of the mass centre at low $k_s$ values confirm the increased macropore infiltration and penetration of solutes

through them to greater depths (cf. Figure 7).

Moreover, the temporal development of the concentrations is similar for all macropore diameters with just marginal differences arising shortly after the rainfall event (Figure A2). While the macropore diameter has a minor influence in the initial phase, stronger differences occur at the end of the simulation when the residual

water and solute amounts of the fictive surface storage have finally infiltrated. Thus, mainly at the end of the simulations the influence of the macropores on the infiltration and the macropore-matrix mixing processes are remarkable, because the storage volume of the preferential flow domain is small and hence it can only collect small amounts of water and solutes in relation to the matrix domain. The centres of mass corroborate the results of Figures 9a, b in the way that the big macropores have the tendency to transport more solute masses into the

subsoil.

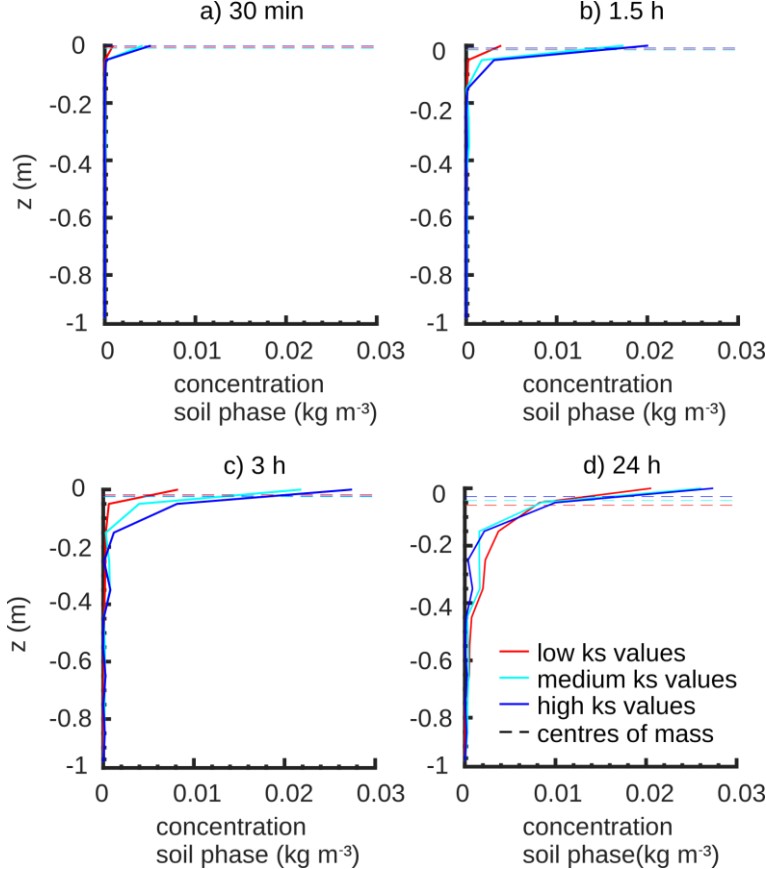

**Figure A1.** Time series of bromide tracer concentration profiles and centres of mass at different $k_s$ values during the rainfall event (**a+b**), shortly after it (**c**) and at the end of simulation (**d**).

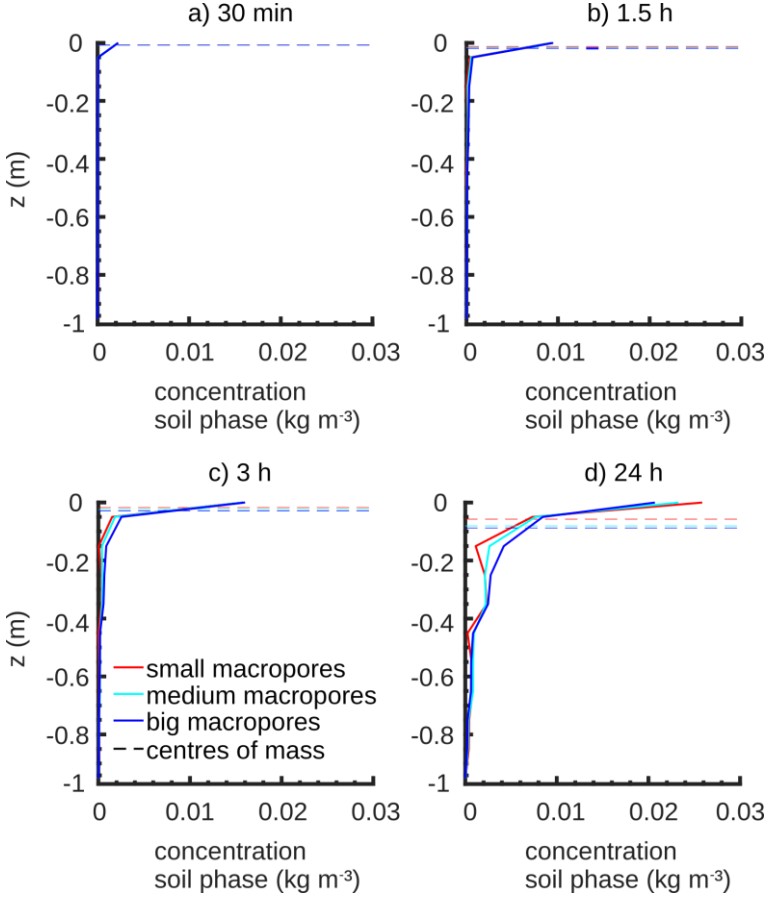

**Figure A2.** Time series of bromide tracer concentration profiles and centres of mass at different macropore diameters (*dmac*) during the rainfall event **(a+b)**, shortly after it **(c)** and at the end of simulation **(d)**.

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

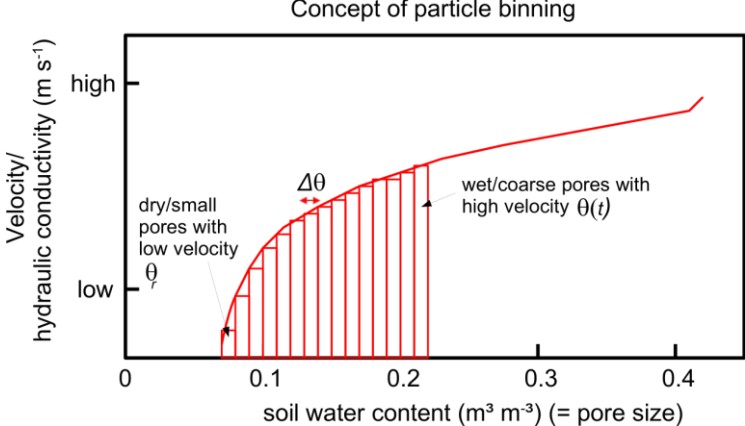

**Figure 1.** Concept of particle binning. All particles within a grid element are subdivided into bins (= red rectangles) of different pore sizes. Dependent on their related bin the particles travel at different flow velocities.

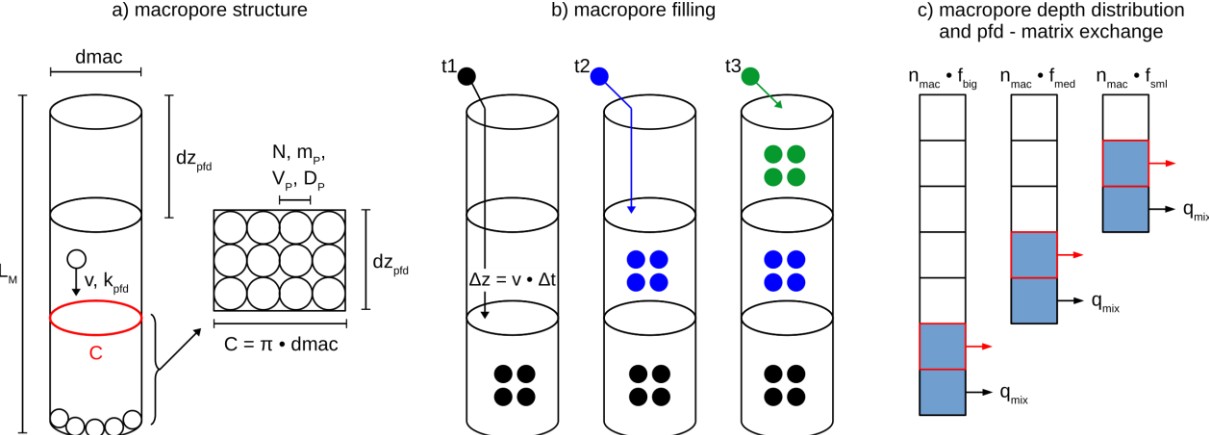

**Figure 2.** Conceptual visualization of **(a)** the macropore structure and cubic packing of particles in the rectangle of a cut open and laid-flat grid element cylinder (cf. section 2.3.1), **(b)** the macropore filling with gradual saturation of grid elements, exemplarily shown for three points in time (t1-t3) whereby at each time new particles (differently coloured related to the current time) infiltrate the macropore and travel into the deepest unsaturated grid element (cf. section 2.3.3) and **(c)** the macropore depth distribution and diffusive mixing from macropores into matrix (cf. section 2.3.4).

**Table 1.** Simulation and tracer experiment parameters (average values) as well as soil hydraulic parameters after Schäfer (1999) at the sites 23, 31, Spechtacker and 33. Where $k_s$ is the saturated hydraulic conductivity of the matrix, $\theta_s$ the saturated soil water content, $\theta_r$ the residual soil water content, $\alpha$ the inverse of an air entry value, $n$ a quantity characterizing pore size distribution, $s$ the storage coefficient and $\rho_b$ the bulk density. In general, all these observable parameters can be freely adjusted in our model and are hence independent from other variables. All other calculated parameters presented in the text are dependent on these observable parameters.

| Parameter | Site 23 | Site 31 | Spechtacker | Site 33 |
|---|---|---|---|---|
| *Irrigation duration (hh:mm)* | 02:10 | 02:10 | 02:30 | 02:20 |
| *Irrigation intensity (mm h$^{-1}$)* | 10.36 | 10.91 | 11.1 | 9.7 |
| *Br-concentration of irrigation water (kg m$^{-3}$)* | | | 0.165 | |
| *Recovery rate (%)* | 77 | 76 | 95 | 96 |
| *Initial soil moisture in 15 cm (%)* | 20.5 | 25.3 | 27.4 | 22.3 |
| *Initial soil moisture in 30 cm (%)* | 25.3 | 15.9 | - | - |
| *Initial soil moisture in 45 cm) (%)* | 28.1 | 13 | - | - |
| *Initial soil moisture in 60 cm (%)* | 29.6 | 13.4 | - | - |
| *Simulation time t (s)* | | 86400 (=1 Day) | | |
| *Time step $\Delta t$ (s)* | | 120 | | |
| *Particle number in matrix (-)* | | 1 Mill. | | |
| *Particle number in pfd (-)* | - | - | 10 k | 10 k |
| *Soil type* | Calcaric Regosol | Calcaric Regosol | Colluvic Regosol | Colluvic Regosol |
| $k_s$ *m s$^{-1}$* | $0.50 \cdot 10^{-7}$ | $0.50 \cdot 10^{-6}$ | $2.50 \cdot 10^{-6}$ | $2.50 \cdot 10^{-6}$ |
| $\theta_s$ *(m³ m$^{-3}$)* | 0.44 | 0.44 | 0.4 | 0.4 |
| $\theta_r$ *(m³ m$^{-3}$)* | 0.06 | 0.06 | 0.04 | 0.04 |
| $\alpha$ *(m$^{-1}$)* | 0.4 | 0.4 | 1.9 | 1.9 |
| *n (-)* | 2.06 | 2.06 | 1.25 | 1.25 |
| *s (-)* | 0.26 | 0.45 | 0.38 | 0.38 |
| $\rho_b$ *(kg m$^{-3}$)* | 1300 | 1300 | 1500 | 1500 |
| *nmac (-)* | - | - | 16 | 46 |
| *dmac (m)* | - | - | 0.005 | 0.006 |
| *Grid element length in pfd dz$_{pfd}$ (m)* | - | - | 0.05 | 0.05 |
| *mac. big (m)* | - | - | 1 | 1 |
| *mac. med (m)* | - | - | 0.8 | 0.6 |
| *mac. sml (m)* | - | - | 0.5 | - |
| $f_{big}$ *(-)* | - | - | 0.13 | 0.35 |
| $f_{med}$ *(-)* | - | - | 0.19 | 0.65 |
| $f_{sml}$ *(-)* | - | - | 0.68 | - |

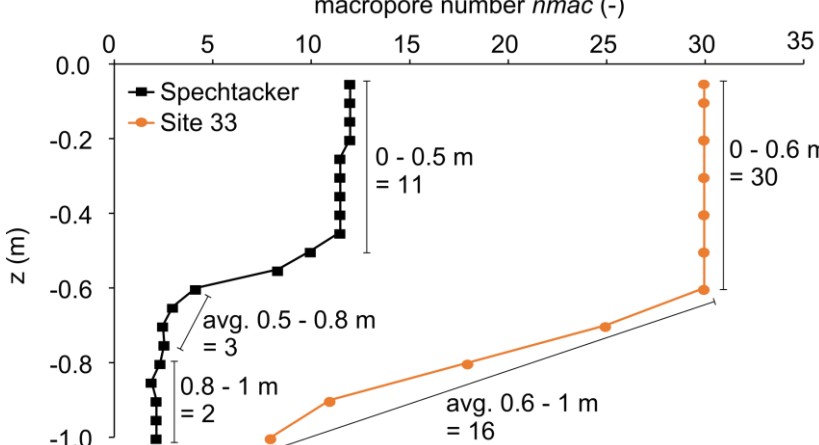

**Figure 3.** Distribution of macropore numbers with an average diameter of 5 mm (Spechtacker) and 6 mm (site 33) along the vertical soil profiles at the two study sites. The arrows highlight the derivation of the macropore numbers in different depths (cf. section 3.2), whereby "avg." means that in these depths the macropore numbers are averaged because there was no clear macropore pattern observed.

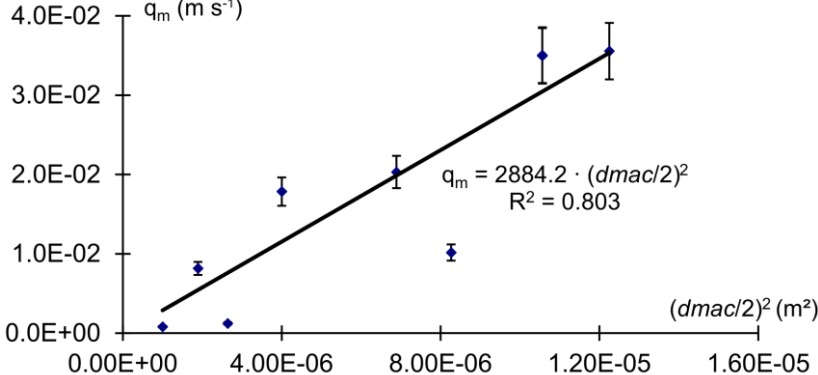

**Figure 4.** Linear regression of the flux rate within the macropore on the macropore radius ($dmac$/2) at the study site Spechtacker (edited figure, adopted from Zehe and Flühler, 2001b). This relation was derived from measurements of saturated flow through undisturbed soil columns containing worm burrows.

**Table 2.** Parameter ranges of the sensitivity analyses and configurations of macropore depth distribution and distribution factors (cf. Figure 10).

| Parameter | Value range | | | |
|---|---|---|---|---|
| $k_s$ (m s$^{-1}$) | $10^{-6} - 10^{-5}$ (step: $1 \cdot 10^{-6}$) | | | |
| $dmac$ (m) | $0.0035 - 0.008$ (step: 0.0005) | | | |
| $nmac$ (-) | $11 - 20$ (step: 1) | | | |
| **mac. depth distr. config.** | **1** | **2** | **3** | |
| *mac. big* (m) | -1 | -0.8 | -0.6 | |
| *mac. med* (m) | -0.8 | -0.6 | -0.4 | |
| *mac. sml* (m) | -0.6 | -0.4 | -0.2 | |
| **distr. factors config.** | **A** | **B** | **C** | **D** |
| $f_{big}$ (-) | 0.13 | 0.3 | 0 | 0.5 |
| $f_{med}$ (-) | 0.19 | 0.3 | 0.2 | 0.3 |
| $f_{sml}$ (-) | 0.68 | 0.3 | 0.8 | 0.2 |

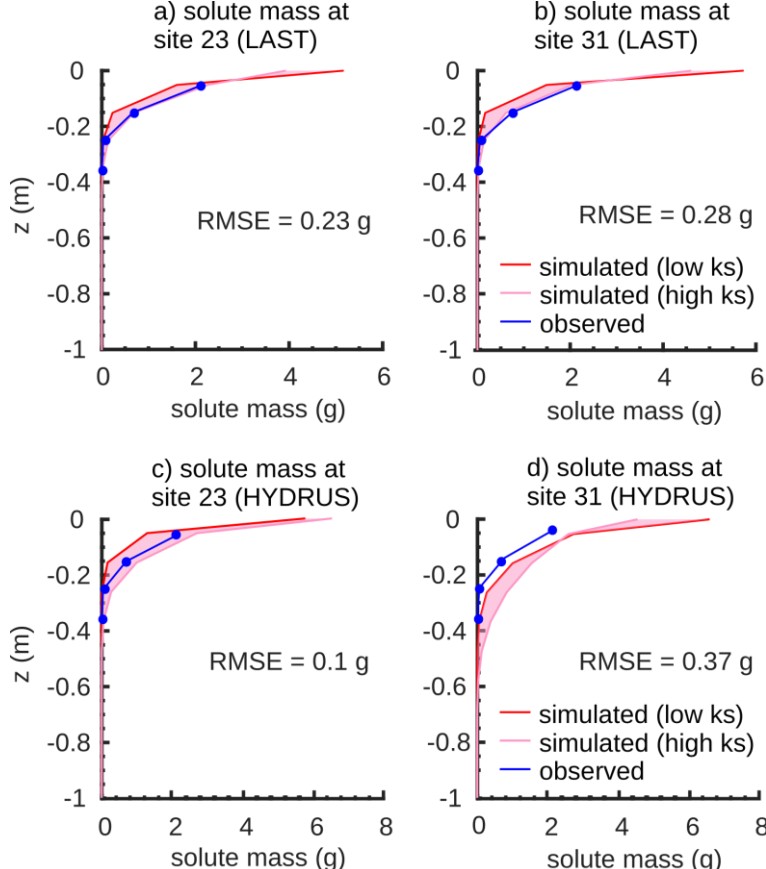

**Figure 5.** Final simulated and observed vertical bromide mass profiles of the matrix at the two well-mixed sites 23 + 31 **(a+b)** with RMSE values simulated with the LAST-Model. In comparison, final simulated and observed vertical bromide mass profiles at the two well-mixed sites 23+31 **(c+d)** with RMSE values simulated with HYDRUS 1-D. The rose shaded area shows the uncertainty area of measured $k_s$ values.

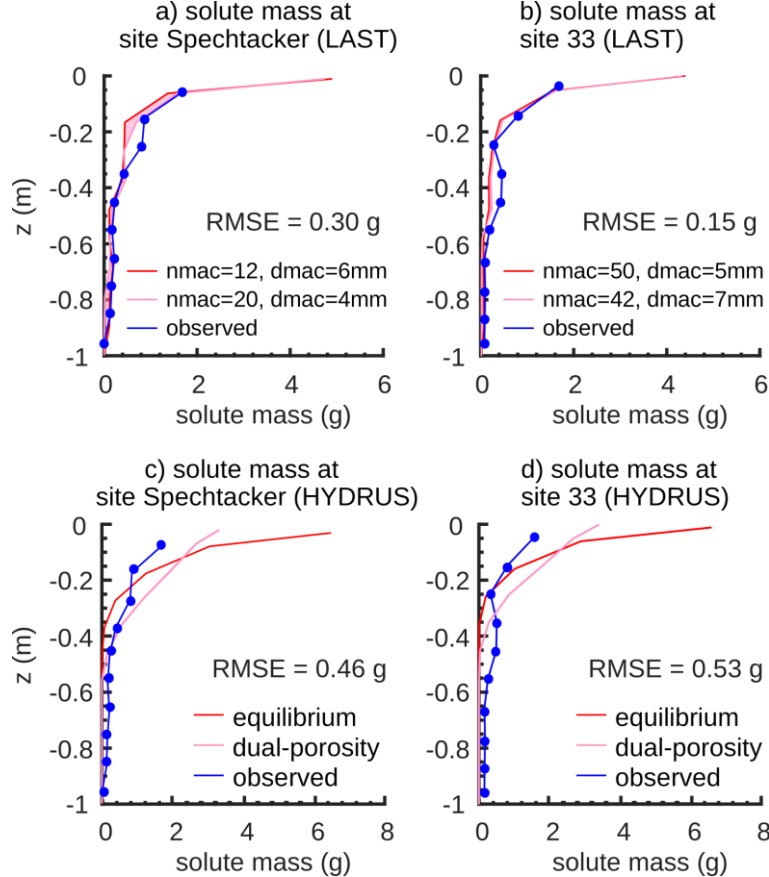

**Figure 6.** Final simulated and observed vertical bromide mass profiles of the matrix at the two preferential flow sites Spechtacker + 33 **(a+b)** with RMSE values simulated with the LAST-Model. The rose area shows the standard deviation of measured macropore numbers and diameters from the mean values at site Spechtacker (*nmac* = 16, *dmac* = 5 mm) and site 33 (*nmac* = 46, *dmac* = 6 mm) (cf. Table 1). In comparison, final simulated and observed vertical bromide mass profiles at the two preferential flow sites Spechtacker + 33 **(c+d)** with RMSE values simulated with HYDRUS 1-D. The rose mass profile is simulated with a dual-porosity approach to account for preferential flow (cf. section 3.3) and for comparison, the red mass profile is simulated with an equilibrium approach.

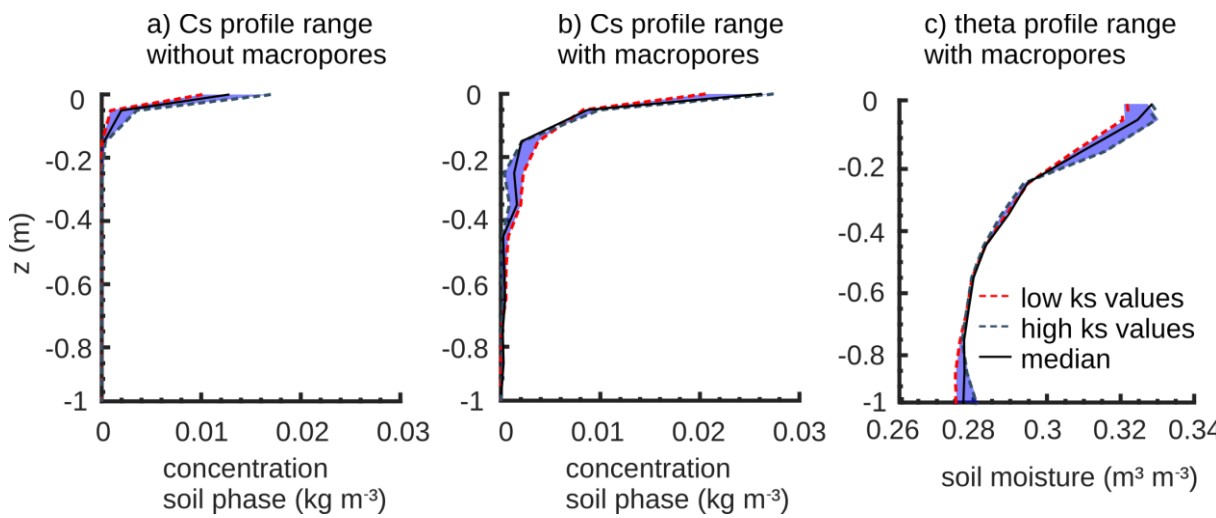

**Figure 7.** Final simulated bromide concentration (*Cs*) and soil moisture (*theta*) profiles of the soil matrix **(a)** without and **(b+c)** with macropores at different $k_s$ values. The blue area shows the possible range of simulated profiles with different $k_s$ values.

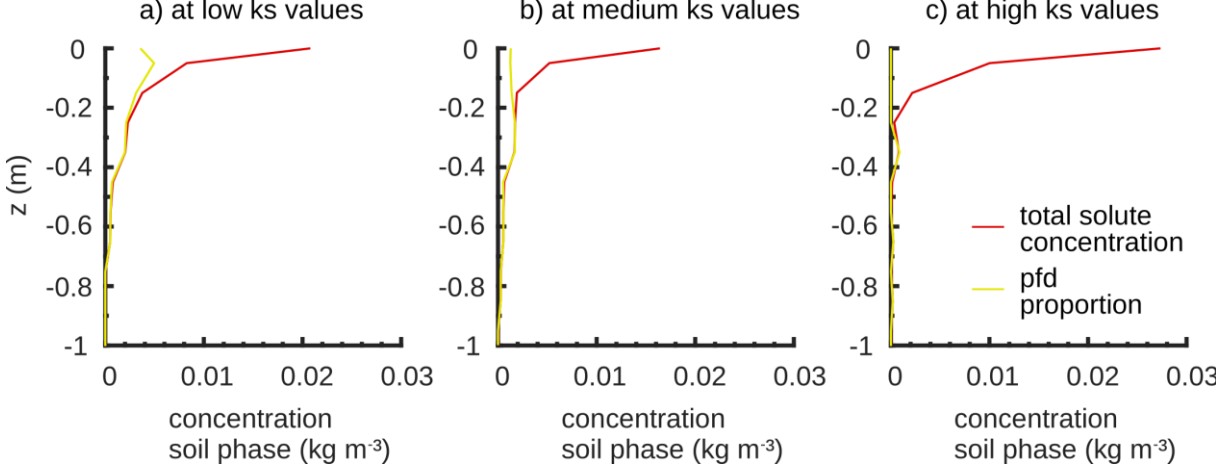

**Figure 8.** Final bromide concentration profiles at **(a)** low, **(b)** medium and **(c)** high $k_s$ values and the proportion of solutes which originates from the macropores.

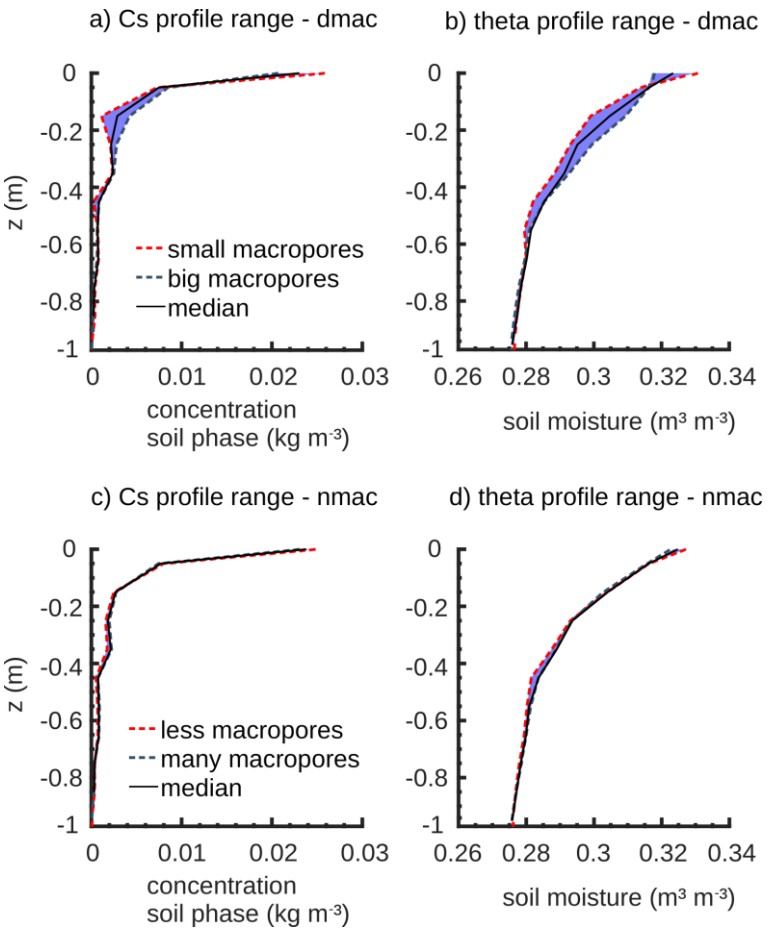

**Figure 9.** Final simulated bromide concentration (*Cs*) and soil moisture (*theta*) profiles of the soil matrix at different macropore diameters (*dmac*) **(a+b)** and macropore numbers (*nmac*) **(c+d)**.

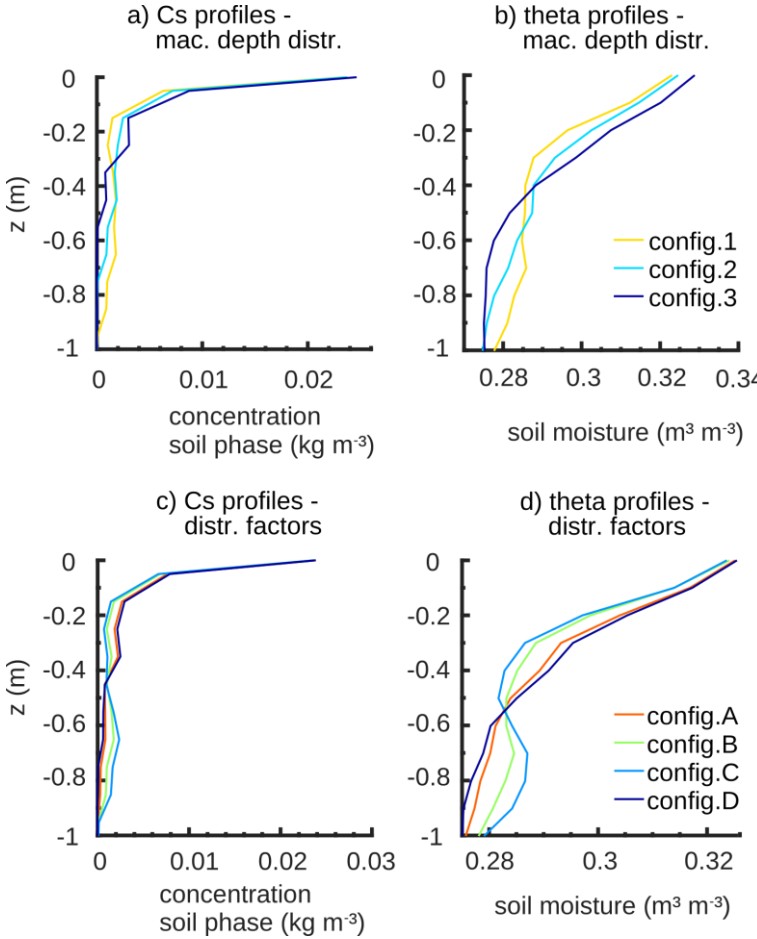

**Figure 10.** Final simulated bromide concentration (*Cs*) and soil moisture (*theta*) profiles of the soil matrix at three different macropore depth distribution configurations **(a+b)** and at four different distribution factor configurations **(c+d)** (cf. Table 2).