# Peer review of "Simulating preferential soil water flow and tracer transport using the Lagrangian Soil Water and Solute Transport Model"

_Hydrology and Earth System Sciences, 2019_

## Referee Comment (RC1) · Anonymous Referee #1 · 6 May 2019

Modelling infiltration processes and the associated tracer transport in a porous media is a challenging task for the possible presence of strong spatial heterogeneities in the hydraulic parameters that characterize the soil. In particular, the presence of macropores can lead to preferential flows that are difficult to retrieve with traditional Richard-based models. The authors propose a new approach to improve the simulation results of a Lagrangian 1-D model introduced by Zehe and Jackisch 2016. The new LAST-model partitions the precipitation into two domains, a 'slow' domain describing the infiltration in the soil matrix, and a 'fast' domain constituted by a collection of marcopores. There are several parameters that describe these macropores, and the more relevant are the three possible depths of the macropores, the diameter, the saturated hydraulic con-

ductivity and the number of macropores at the different depths. The solute masses are converted into a large number of particles (in this case two milions) for the Lagrangian simulation of the infiltration problem.

To prove the efficacy of the model, the Authors show that the LAST model outperforms the previous Lagrangian model in a real infiltration example. Finally a sensitivity analysis shows the stability of the model and the possible changes when varying different parameters.

Although the problem is of relevance for many HESS readers, I think that more results are required to support and validate the proposed method. In fact, the pfd adds several degrees of freedom for the description of a 1-D infiltration problem, and only one infiltration test is not sufficient to prove its efficacy and the limitations with respect other methods. Details regarding the calibration of the model parameters are missing. Moreover there are many points that need to be better explained in the description of the method, as detailed in the minor comments. For these reasons I recommend major revision.

**MAJOR COMMENTS**

1. The proposed model is introducing a large number of additional parameters that cannot be directly related to physical properties of the soil and that require adequate calibration. I think that using 16 parameters to retrieve 10 data points (Figure 3c) might introduce a strong over-parametrization of of the model. Thus, additional examples of application of the model to real data are needed, including calibration procedures and measures of goodness of fit. In particular, a fair validation would be to compare LAST with a 1-D Richards-based model that considers a simple soil heterogeneity (e.g., hydraulic parameters changing in two or three layers of the domain).

2. Beside calibration, I find very difficult to apply LAST to different infiltration settings. For example, if deeper domains or longer durations of the experiment are considered, would it be always sufficient to have three classes for the length of the macropores? Why not having the classes of macropores evenly spaced along the domain? Moreover, the sensitivity of model results to the number of macropores is very low. Is it possible to consider just one macropore, and consequently adapting its diameter and the diffusion fluxes with the soil matrix?

3. The authors present three real infiltration tests, but compare the new LAST model with respect to the previous infiltration model in only the third example. Why not applying the LAST model also to the other two infiltration tests? Can you please show that proper calibration of the LAST model is suggesting to not consider the pfd component in those tests?

**MINOR COMMENTS**

1. Page 2, line 20: please insert a reference for the 'Double domain model'.

2. Page 5, line 10: the caption of Figure 2 is not sufficient to understand the figure. The figure should be better explained in the text. In particular, which is the relation between the grid element and the macropores?

3. Page 5, line 15: also the concept of cubic storage is really vague from the text and the figure and it is not in agreement with the cylindrical shape of the macropores.

4. Page 5, line 33: in Case 3 the accumulated water should create a ponding volume for both the soil matrix and the macropores. Why this is not taken into account in equations (3) and (4)?

5. Page 5, line 36: $m_{matrix}$ and $m_{pfd}$ described in equations (3) and (4) should be the infiltration capacities, not the mass of water that infiltrates as stated in line 36.

6. Page 6, eq. 3: at my understanding, this equation is approximating the infiltration capacity for the first grid element. Why does it involve the potential gradient in the second grid element?

7. Page 6, eq. 4: the power 2 should be outside the parenthesis.

8. Page 6, lines 20-23: does this mean that the time step changes at each temporal iteration? In fact the deepest unsaturated grid element changes in time. Do the macropores fill at the same speed? From what I understand at lines 10 and 11, the water reaches different depths during the same time step, depending on the depth of the pdf. I think this should be clarified.

9. Page 6, line 25: what is the boundary condition at the bottom of the macropores? From this description, the model can handle only no-flow condition, which is a big limitation.

10. Page 6, line 14: what does the term 'coupled' means here? Does this mean that the water in these grid elements entered the system at the same time (and thus have the same tracer concentration)?

11. Page 7, line 10: why three depths? This seems a very arbitrary choice without a real physical meaning.

12. Page 7, line 15: how can the diffusive water flow be simultaneous? The water in the small macropores reaches the deepest unsaturated level much faster than the water in the big macropores, thus it should start the diffusive flow before.

13. Page 7, line 20: Why are the Authors using a harmonic mean in (6) and an arithmetic mean in (3)?

14. Page 7, line 24: it is still not clear what are $C$ and $D_M$ and their meaning is the opposite of what is defined in the caption of Figure 2.

15. Page 8, Lines 4, 7: what is the depth of the soil samples? Please specify also the initial soil saturation used in the model.

16. Page 9:, eq 7: how was this coefficient computed?

17. Page 9, section2.4.3: the proposed macropore structure has many degrees of freedom. Is it possible to calibrate / validate such a model with infiltration measurements?

18. Page 9, line 20: Which kind of sensitivity analyses is performed? Sensitivity of which output of the model? From table 2 I understand that the parameters used in the sensitivity analysis are evenly spaced in the parameter space. Usually in MC-approaches the parameters are randomly selected from the parameter space.

19. Page 9, lines 31 – 33: this part should go in the discussion.

20. Page 10, line 1: This sentence is not clear here. I suggest to describe the observations (and the possible difference with the mode outputs) in the methods section. Please provide more information about how these observations are obtained. The real process is three-dimensional. How are these concentrations obtained? Are they an average of the concentrations in different layers?

21. Page 10, line12: change 'suggest' with 'suggests'

22. Page 11 , line 3: please specify which are the three values of $k_s$ considered

23. Page 11, Section 3.3.2: please discuss why in figures 7a and 8d the concentration increases between depths -0.15 and -0.4 for all the macropores diameters considered.

24. Page 12, line 30: This sentence in not correct: to prove this sentence, the sensitivity analysis should be performed by perturbing the parameters of the real-case

experiment. However the parameters of the real-case experiment are not among the ranges considered in the sensitivity analysis.

25. Page 12, lines 31-33: this modelling detail should be specified in the model setup. Which is the computational cost of the model when using 2 million particles?

26. Page 13, line 18: From figure 5, the sensitivity of the results with respect the considered variation of $k_s$ is quite small. What are the differences when changing $k_{pfd}$?

27. Figure 2, caption: the saturated hydraulic conductivity of a macropore here is indicated with $k_s$, while in the main text (page 5, line 8) it is indicated with $k_{pfd}$. I think the same notation should be used for these two variables. From the figure I am not sure to understand the difference between the diameter of macropore ($D_M$) and the circumference of a grid element ($C$). Why the length of a grid element is expressed as $dz(\Delta z)$ ?

28. Figure 2b: this figure is really not clear and not well explained in the caption. Do the three cylinders correspond to three different time-steps? The times should be better indicated in the figure and the caption should describe what is happening in the three steps.

29. Table 2: please consider using a parameter range that covers the parameters used in the Spechtacker test. Not only it is important to see the sensitivity of the model in this test, but I think the results obtained for the Spechtacker are quite interesting, having a very deep infiltration.

30. References: please check the references: Zehe and Bloshl in not in the correct place. Sometimes there is an 'and' before the last author, sometimes not.

---

## Referee Comment (RC2) · Anonymous Referee #2 · 7 May 2019

In this work, the authors extend the model of Zehe and Jackish (2016) considering transport and linear mixing of solute in the soil matrix and introducing the concept of preferential flow through macropores. They used a Lagrangian approach in accordance with the Zehe and Jackish, modelling water infiltration in an effective 1-D soil domain. They take into account preferential flow in macropores, partitioning water into two domains (macropores and matrix), similarly to a double continuum model. They define a preferential flow domain, modeling infiltration into the macropores and the matrix and diffusive mixing from macropores into matrix. They perform a sensitivity analysis for some model parameters and they apply the model to a tracer experiment. In my opinion the object of this work is of interest for Hydrology and Earth System

[Figure]

Sciences Journal, however I suggest a major revision. The model is characterized by a large number of parameters, that, in my opinion, are not properly described. As a consequence, it is not clear, from my point of view, how the model has been applied to the only tracer experiment provided and how these parameters could be linked to some physical property. Sensitivity analysis has been shown only for few parameters, considered the most relevant, and only few words has been spent for some other parameters (e.g. the distribution factor: fbig, fmd fsml and the matrix potential). Some statements are not supported by evidences and some results are not commented, in the following my major and minor comments.

MAJOR COMMENTS

- As mentioned in the Introduction and in the Conclusions this model is similar to a double-continuum one. In my opinion, it would be interesting if you could elaborate this similitude, ideally linking the parameter of your model with the ones of a classical double approach (with appropriate references)

- Pag.4, line 23: here I am a bit confused about how do you compute the diffusive mixing: it is not the entire solute mass in a grid element given by the mass of all the present water particles?

- Pag.4, line 26: do you have some criteria to define a "sufficiently fine" grid?

- Pag.5 line 10: could you please list all the parameters of the model and do not only refer to Figure 2 in order to better clarify how many parameters the model counts? Is the pfd characterized by the 17 parameters given in the caption of Figure 2? Is the macropore diameter "dmac" (in the text) equal to "D_M" (in Figure 2 and in Section 2.3.i)? In Table 1 we have 16 parameter for the soil description (7 for the soil type and 9 for the macropores domain) + 8 for the experiment conditions + 4 for the numerical implementation.

- Pag. 6, line 6: how is the matrix potential gradient between the first two grid elements

computed?

- Pag. 7, line 11: why does your model "generally" divide the total amount of macropores into 3 parts? Could you please explain the meaning and the effect distribution factor?

- Pag. 7, line 20: why do you use the harmonic mean to compute a sort of "effective" hydraulic conductivity? Usually, the effective hydraulic conductivity for heterogeneous media in parallel configuration is computed as the arithmetic mean of the 2 conductivities ... is your model sensitive to this choice ?

- Eq(7): is the number "2884.2" result of a calibration? Could you please provide some details about it? Have you calibrate some parameter of your model to fit the experimental data? Could you please provide some details about how to use the model to interpret experimental data?

- Pag.9, line 21: in my opinion, the way you describe your sensitivity analysis is a bit vague .. how do you conclude that ks, dmac (=Dm?), nmac are the most sensitive parameters ? How do you conclude that ks is "probably the most sensitive parameter" (Pag.9, line 31) ?

- Pag. 10: Result section: I am sorry, but for me it is not clear how do you select the parameters of your model to simulate the tracer mass in the respective depths, could you please state more clearly which observables you had, which parameters you compute from measurements etc.

- Pag. 10, line 23: do you have an explanation about the greatest difference in profiles between 0.15 and 0.35m depth?

- Pag. 10, line 32: here, as in Figure 9, it looks that the results are sensitively depending on the Configuration (1,2,3) . . . there is a way to parametrize the different configurations in order to study and quantify the sensitivity of the model to the different configurations?

- Pag.12, line 28: here, you conclude that macropore-matrix exchange should be modelled deriving an "effective conductance", even if it is the first time you introduce this term. I suppose you refer to the coefficient in Eq.6, but I would specify it.

- Pag. 13, line 3: I agree that "further field experiments on a variety of differently structured soil is necessary", however, from my point of view, it is not clear how do you parametrize these differently structured soils as well as do you parametrize the spatial heterogeneity of the macropores network (Pag. 13, line 35)

- Pag. 13, line 13: in my opinion it is not so straightforward how do you transfer the concept of cubic particle storage and hydraulic radius to any kind of macropore geometry.

MINOR COMMENTS

- Eq. (1): please check this equation. I suppose a "+ $z_i(t)$" after the equal is missing and the format is different from the other equation in the manuscript.

- Pag. 4 line 5: could you provide some details about the soil water retention curve used to compute the diffusivity from the hydraulic conductivity?

- Pag. 4, line 6: I guess that Z is a random uniformly distributed number "between 0 and 1";

- Pag.4, line 21: could you please provide some details about the numerical implementation of the model (e.g. programming language etc)?

- Pag. 6, line 4: typo: please write consistently $k_{m1}$ with Eq.(3) as well as $n_{mac}$ introduced in Pag.5, line 6.

- Pag 6, line 8: is the simulation time step "dt" or "Delta t"?

- Pag 7, line 22: please correct a typo: "matric" potential

- Pag. 12, line 33: Here you say that you need at least two million particles, but I suppose the minimum number of particles you need is proportional to the observation

area, isn't it?

- Pag. 13 line 38: you conclude that your model provides high computational efficiency with short simulation times, could you please provide further details?

———————————————

---

## Referee Comment (RC3) · Anonymous Referee #3 · 8 May 2019

A Lagrangian modelling method developed in Zehe and Jackisch (2016) have been extended to simulate solute mixing in soils with macropores which act as preferential flow pathway. It was tried to check validity and applicability of the extended model for replication of field data and solute transport processes. The topic is of significant importance for geoscientists and hydrogeologists, considering the challenges of using existing models for simulation of mixing in heterogeneous soils with micropores which have been discussed sufficiently in the introduction section of the paper. The proposed model needs a specific parameters describing the characteristics of the pore scale structure of the soil such as dimension of macropores. A strategy for extending the modelling method has been described smoothly and appropriate level of details have

been presented. However, it should have been verified using simple benchmark examples to check reliability of the method before any effort for its validation with field data.

Major comments: 1) The modelling strategy have been proposed to overcome challenges related to dual domain models, however, there no quantitative comparison between the dual domain methods proposed in Seven and Germann, (1981) or Nezhad et all (2010), and the model proposed by authors in this manuscript. A further analyses is required to compare the results achieved from the extended work and the original LAST model as well as results that can be achieved via dual domain theory. These quantitative comparisons are required, particularly, for clarification of discussions in lines 25-30 if the page 12.

2) Some new parameters have been introduced in the new model, which may not be physically measurable such as dimension of the micropores and considering the authors effort for simulation of field data, it has not been proposed/specified how values of these parameters can be identified.

3) Discussion regarding computational efficiency of the proposed model has not been presented sufficiently, and for example in page 13 line 39 duration of simulation has been presented without identifying which machine have been used and also duration of simulation with other possible model have not been compared. With our such complete comparisons, discussions on efficiency of the method would not add any scientific knowledge to the readers.

4) Some of the results presented in the paper are obvious and do not need complex modelling methods to be implemented. For example discussions presented in page 12 lines 15-20, can be achieved using other methods and perhaps developing proposed model was not required to understand these. Perhaps if authors compare their results with other results achieved using other methods which capture the effects of macropores, more valuable finding will be presented. Authors should make the results section

more focused on the capacity of new strategy used for modelling micropores and their interactions with soil matrix.

Minor comments: 1) Simulation domains have not been explained sufficiently in the text, and mainly some figures have been presented which are not enough to understand the problem being simulated.

2) A complete description of boundary conditions and initial conditions for simulation domains are required.

3) discussion on time step in page 6 lines 20-25 is vague and needs to be clarified. It will be helpful that author visualise the discussion and king it more understandable.

)If I understood correctly LAST model is the same as the model developed by Zehe and Jackisch (2016). I suggested that author call it as their model or the model developed by Zehe and Jackisch (2016), i.e., rewrite lines 9-11, I suggest "Our LAST-Model (Lagrangian Soil Water and Solute Transport) developed by Zehe and Jackisch (2016) relies on the movement of water particles carrying a solute mass through the soil matrix and macropores. We advance this model by two main extensions: a)..."

—————————————————

---

## Short Comment (SC1) · 15 May 2019

*A note upfront from the submitting person: This review was prepared by Sandra Werthmüller and Jasmin Kesselring, both master students in Earth System Science at the University of Zurich. The review was part of an exercise during a second semester master level seminar on "the biogeochemistry of plant-soil systems in a changing world", which I organize. We would like to highlight that the depth of scientific knowledge and technical understand- ing of these reviewers represents that of master students. We enjoyed discussing the manuscript in the seminar, and hope that our comments will be helpful for the authors.*

[Figure]

—- Sternagel et al. (2019) developed a model to simulate preferential water flow and tracer transport in macroporous soil. As a starting model, they have used the Lagrangian model of Zehe and Jackisch (2016) and added two extensions to the model. Firstly, they added a solute concentration (C) to each particle so that solute transport can be simulated. Secondly, the inclusion of a preferential flow domain allows to simulate the influence of macropores on water and solute dynamics. To evaluate the model Sternagel et al. (2019) used data of an experiment that was done in the Weiherbach catchment by Zehe and Flühler (2001). They concluded that their simulation of solute transport under well-mixed conditions corresponds generally well with the observed data from Zehe and Flühler (2001). The same is true for the preferential flow domain. The additional sensitivity analysis that they conducted shows that the conductivity (ks) of the soil has a major impact on the infiltration of the water.

General comments: In general, we think that the manuscript has a good structure and one can follow the development of the model the way it is described in the paper. However, we think that the introduction is slightly too long compared to the rest of the manuscript.

For us as beginners in the field, it is hard to understand why your model is innovative. Could you explain at the beginning of the paper what makes your model innovative compared to others in the field? And how your work is embedded in the broader work of soil water modelling? We understand that the paper is about discussing the development of a new model and is thus theoretical. However, we think a more practical description of the use of the model would be nice. For instance: For which studies is this model a must have addition? We also think that the model would have to be compared to more than one practical study to fully be called a valid model. There are some additional points we find unclear:

Page 3 Line 36 ff: How is the number of bins i and the subdivision into N bins defined? What exactly is the difference between those two and how do you choose the 'perfect' number of bins?

Page 4 Line 30-33: Here, you list four subchapters that will follow in the next paragraph. Why not name the actual subchapters according to this list?

Page 9 Line 31ff: You already start the interpretation of results, why not in the dedicated section (discussion and conclusion)?

The layout of your references makes it hard to differentiate references. We also noticed that a lot of citations and references you used are from the same authors. We were wondering, if there are other scientists that are working on the same problem to which you could compare your results with.

—- Detailed comments: The abbreviation for confer is cf. not c.f. It is used inconsistently in the manuscript.

Page 1 Line 34: become a major issue (change an to a)

Page 4 Line 24ff: This sentence is a bit difficult to understand. Maybe make two sentences e.g. ...corresponding to the molecular diffusion coefficient. Additionally, this needs to be smaller than. . .

Page 6 Line 4: $k_{m1}$ or $k\_m1$ with a subscript 1 as in the formula above?

Page 8: Has unnecessary empty space

Page 9 Line 21ff: In this sentence you suggest that the parameter hydraulic conductivity of the matrix $k_s$, diameter of macropores $d_{mac}$ and the amount of macropores $n_{mac}$ are the most sensitive for the model behaviour and simulation results. Please elaborate why and give a reference for it.

Page 9 Line 24ff: In this paragraph you mentioned different configurations for depth distribution and distribution factors. They have the same numbers, which is confusing and makes the text hard to understand. If possible, clarify the difference between depth distribution and distribution factors.

Page 13 Line 37 ff: You mention that your model is highly computational efficient and

with a short simulations time (about five minutes). How does this short simulation time compare to other similar models? Could you give a reference time? And could you explain how this new model increased computational efficiency?

Figure 1: Why are pore size and soil water content equal to each other? (x-axis) Maybe mention in the figure caption how the bin width is calculated.

Figure 2: In line 3 of the caption: describe DM, LM, dz separately like the other parameters and not as a group. We do not understand what figure b) means. What do the different colours stand for? Describe it better in the text where you reference it as well as in the figure caption.

Figure 3/4: Is the coloured in area the uncertainty range? Are these different parameters in figures 3 and 4 or why do they have different colours? For us the graphics are also a bit small which makes it difficult to read them. It would be better if the graphics were a bit bigger.

Figure 9: In all four plots use the same colour for the same configuration number. This makes it easier to see the influence of the different factors on the configurations.

---

## Author Comment (AC1) · 20 May 2019

**Response to Comments of Anonymous Referee #1**

On behalf of all co-authors I sincerely thank the Anonymous Referee #1 for his thoughtful and detailed assessment of our work.

Major Comments

**R1:** The proposed model is introducing a large number of additional parameters that cannot be directly related to physical properties of the soil and that require adequate calibration. I think that using 16 parameters to retrieve 10 data points (Figure 3c) might introduce a strong over-parametrization of of the model. Thus, additional examples of application of the model to real data are needed, including calibration procedures and measures of goodness of fit. In particular, a fair validation would be to compare LAST with a 1-D Richards-based model that considers a simple soil heterogeneity (e.g., hydraulic parameters changing in two or three layers of the domain).

**AS**: Indeed, the pfd is mainly characterized by 9 parameters (in this case the macropore lengths, diameter, distribution factors, grid element length). The other characteristics like the volume, lateral area etc. depend on those parameters and the flow rate depends on the macropore diameter (compare Fig. 1 of this manuscript, note that this relation was derived by measurements of saturated flow through undisturbed soil columns, which were centered around worm burrows). Hence, at least for worm burrows, the depth distribution and the diameters are observable (compare Fig. 2 of this manuscript) and not arbitrary calibration factors. This will be better explained in the revised manuscript by an additional chapter presenting the model database with figures and showing which observables we had and how we obtained our pfd parameters from these.

[Figure]

Figure 1: Linear Regression to evaluate the relation of macropore radius and flux rate within the macropore the study site Spechtacker (Zehe et al. (2001)). This relation was derived from measurements of saturated flow through undisturbed soil columns containing worm burrows.

[Figure]

Figure 2: Patterns of dye tracer (a+d) and worm burrows as well as the measurement of distribution, lengths and diameters of those macropores in different horizontal layers (d) at the study site Spechtacker (taken from van Schaik et al. (2014)).

Furthermore, other double domain models also rely on extensive parametrization. In case these models rely on the kinematic wave theory, these parameters are for instance the maximum flow rate in the macropore system, the exponent characterizing how the actual flow rate increases with saturation of the macropore domain, and an exchange length to calculate potential gradients driving macropore matrix exchange. The latter two parameters need to be calibrated as well.

Moreover, we generally agree that a comparison with a Richard solver is interesting. In case of pure water flow this has already been done by Zehe and Jackisch (2016) who revealed a good accordance of both approaches. And in this particular case, the Richards solver and the particle model had the same amount of parameters, as the diffusivity and the drift parameter of the random walk are derived from the soil water retention and the soil hydraulic conductivity curves. In the revised paper, we plan to additionally test our model against a Darcy-Richards approach, e.g. re-simulation of our three infiltration tests with HYDRUS 1-D and comparison. To this end, please see Figure 3 of this manuscript which shows the results of the simulation of our three infiltration tests with HYDRUS 1-D compared to the results of our LAST-Model.

As you can see, at the well-mixed study sites 23 and 31 HYDRUS 1-D performs well in accordance to the observed values and it is also similar to our simulation results with just slight deviations but which are in the range of uncertainty. In contrast, at the preferential flow study site Spechtacker HYDRUS 1-D with its double-domain approach is not able to simulate well the highly heterogeneous, observed solute mass profile. Here, our model performs much better in comparison. We will discuss these results in our revised paper in more detail.

[Figure]

Figure 3: Solute mass profiles at our three study sites simulated with HYDRUS 1-D (lower part) and compared to the mass profiles simulated with our LAST-Model (upper part)

**R1:** Beside calibration, I find very difficult to apply LAST to different infiltration settings. For example, if deeper domains or longer durations of the experiment are considered, would it be always sufficient to have three classes for the length of the macropores? Why not having the classes of macropores evenly spaced along the domain? Moreover, the sensitivity of model results to the number of macropores is very low. Is it possible to consider just one macropore, and consequently adapting its diameter and the diffusion fluxes with the soil matrix?

**AS:** We assume that a macropore distribution with three different lengths is a sufficient approximation of the observed macropore depth distribution at the study site Spechtacker. Nevertheless, as a variable macropore depth distribution might be observed at other sites, we agree that the model needs to be more flexible in this respect.

Of course it is possible to represent the macropore network by just one big pore. But please note that the macropore diameter is limited in reality, in case of worm burrows usually up to a maximum of 4-5 mm. As we use a linear regression to estimate the flux rate based on the macropore geometry, we restrict the diameters of macropores to a realistic range to avoid extrapolations to unrealistic flow rates (Figure 1 of this manuscript). In an early stage of the development of the pfd we tested the idea of representing the entire macropore network as just one big macropore. In relation to volumes, masses and particle masses this would not make any difference but due to the large diameter of the one macropore the diameter-dependent flux density would be unrealistically high.

Secondly, a large macropore has a different relation of macropore cross section to the perimeter compared to many small macropores. This relation is important to calculate the fraction of particles which contribute to the exchange with the matrix and this was also the reason why we used a more realistic representation of the macropore network with a certain amount of smaller macropores. We will better explain this in the revised manuscript.

**R1:** The authors present three real infiltration tests, but compare the new LAST model with respect to the previous infiltration model in only the third example. Why not applying the LAST model also to the other two infiltration tests? Can you please show that proper calibration of the LAST model is suggesting to not consider the pfd component in those tests?

**AS:** Sorry, if our explanations were unclear and led to misunderstanding. We applied our LAST-Model on all three presented infiltration experiments. We will stress this more properly in our revised paper.

But de facto at the first two well-mixed study sites there are no or just a little active pfd because observed tracer patterns and the excavation of soil profiles did not reveal any considerable macropore network, therefore we assume well-mixed flow conditions without a considerable influence of macropores at these two sites.

We plan to perform a simulation of an additional infiltration experiment at another preferential flow site to provide more comparable model results in our revised manuscript.

Minor Comments

**R1:** Page 2, line 20: please insert a reference for the 'Double domain model'.

**AS:** Thanks, we will add a reference.

**R1:** Page 5, line 10: the caption of Figure 2 is not sufficient to understand the figure. The figure should be better explained in the text. In particular, which is the relation between the grid element and the macropores?

**AS:** We will edit Figure 2 and its caption to make it easier to understand. In general, grid elements are vertical sub-elements of a macropore, similar to the grid elements of the matrix. The grid elements of both matrix and pfd are necessary to create small spatial discretizations for the calculation of the new state variables (soil moisture, solute concentration, hydraulic conductivity) in each time step and in this way to register even slight spatial and temporal alterations of the state variables.

We will add a revised version of Figure 2 and a better explanation to the revised manuscript. Figure 4 of this manuscript gives an idea of the revised figure.

[Figure]

Figure 4 (i.e. Figure 2 of the revised paper): Conceptual visualization of a) macropore structure and cubic packing of particles within the rectangle of a cut open and laid-flat grid element cylinder, b) macropore filling with gradual saturation of grid elements, exemplarily shown for three time steps ($t_1$-$t_3$) whereby in each time step new particles (differently coloured related to the current time step) infiltrate the macropore and travel into the deepest unsaturated grid element c) macropore depth distribution and diffusive mixing from macropores into matrix.

We will shorten the caption and explain the single parameters presented in the figure within the text of the methods section.

**R1:** Page 5, line 15: also the concept of cubic storage is really vague from the text and the figure and it is not in agreement with the cylindrical shape of the macropores.

**AS:** Maybe cubic packing is a better wording, as the macropore is cylindrical but the water particles are spheres. The particle diameter is determined by the stored water mass, the density of water and the number of particles in the pfd. The amount of spheres which can be packed into a macropore cylinder is calculated from the cubic packing. This means that the particles are arranged in the way that the centers of the particles form the corners of a cube. The concept of cubic packing facilitates the calculation of the proportion of particles having contact to the lateral surface of a grid element. The rectangle in Figure 4a (i.e. Figure 2a in the revised manuscript) of this manuscript describes such a lateral surface of a grid element, with the height dz and the circumference C as length, which can be obtained when a macropore grid element is cut open and laid-flat. The number of particles which fit into this rectangle have then contact to the lateral surface.

**R1:** Page 5, line 33: in Case 3 the accumulated water should create a ponding volume for both the soil matrix and the macropores. Why this is not taken into account in equations (3) and (4)?

**AS:** As the infiltration rate into the matrix is based on Darcy's law we are generally able to account for an additional hydrostatic pressure due to a ponded surface. This will indeed increase the infiltration rate into the matrix domain and we will implement this into the model. But given our investigated cases with a precipitation rate of roughly 10 mm/h we suggest only small ponding heights with marginal effect. This might be of relevance when the model is used to calculate double-ring infiltrometer experiments with related great

ponding heights. In other cases, we expect that the water will runoff as overland flow. Up to now this is not within the scope of the model and we will better explain this in the revised manuscript.

**R1:** Page 5, line 36: mmatrix and mpfd described in equations (3) and (4) should be the infiltration capacities, not the mass of water that infiltrates as stated in line 36.

**AS:** Yes and no! As the model works with particles with a discrete mass, the infiltrating fluxes of water ($m^3/(m^2 *s)$) needs to be transferred into a mass to calculate the number of infiltrating particles per time. This is the reason why we present the infiltrating masses in both equations. We will better explain this in the text.

**R1:** Page 6, eq. 3: at my understanding, this equation is approximating the infiltration capacity for the first grid element. Why does it involve the potential gradient in the second grid element?

**AS:** Sorry, for the misunderstanding. The first grid element belongs to the soil surface (z = 0) and the second actually to the first grid element right beneath the soil surface (z = 5 cm). Maybe this is not clear enough within the text. We will clarify this.

**R1:** Page 6, eq. 4: the power 2 should be outside the parenthesis.

**AS:** Absolutely true. Many thanks, we will correct that.

**R1:** Page 6, lines 20-23: does this mean that the time step changes at each temporal iteration? In fact the deepest unsaturated grid element changes in time. Do the macropores fill at the same speed? From what I understand at lines 10 and 11, the water reaches different depths during the same time step, depending on the depth of the pdf. I think this should be clarified.

**AS:** Generally, our model can work with variable time stepping as it is not subject to numerical stability criteria. In fact, we select the time step such that the particle displacement per time step equals the maximum depth of the pfd and subsequently we shift excess particles to the deepest unsaturated grid element. In this way we gradually fill the macropores from the bottom to the top (see Fig. 4b of this manuscript) and this further implies that particles reach the bottom of shallow macropores even faster.

**R1:** Page 6, line 25: what is the boundary condition at the bottom of the macropores? From this description, the model can handle only no-flow condition, which is a big limitation.

**AS:** In this case, we indeed used a no-flow lower boundary. Generally, we agree that it is of course important to allow flow at the lower macropore boundary. This can be achieved by

using the same formula as for the lateral exchange but we have to account for the hydrostatic pressure in the saturated parts of the macropores. We will revise the model accordingly.

**R1:** Page 6, line 14: what does the term 'coupled' means here? Does this mean that the water in these grid elements entered the system at the same time (and thus have the same tracer concentration)?

**AS:** Yes, you are right, and it also means that at the pfd-matrix mixing the diffusive flow from the coupled grid elements happens simultaneously (please, see also our response to your comment below).

**R1:** Page 7, line 10: why three depths? This seems a very arbitrary choice without areal physical meaning.

**AS:** Please see the response to your second major comment as we think we already answered your question there.

**R1:** Page 7, line 15: how can the diffusive water flow be simultaneous? The water in the small macropores reaches the deepest unsaturated level much faster than the water in the big macropores, thus it should start the diffusive flow before.

**AS:** We agree with you, that there might be cases where a temporarily resolved treatment is necessary. In the presented cases and the selected time stepping, particles travel along the maximum vertical depth of the pfd within one time step (dt = max. length/vmak). So, the different arrival times are not resolved in this case due to the high advective velocity in the macropores and the relatively small distances. The difference of arrival times and the saturation velocities of the different macropores can be assumed as marginal.

When assuming the particles travel along the minimum vertical depth of the pfd within one time step (dt = min. length/vmak) it would also have just a marginal effect on the different arrival times as the diffusive flow from the big macropores would then probably start just one time step later due to the general high velocities within the macopores.

**R1:** Page 7, line 20: Why are the Authors using a harmonic mean in (6) and an arithmetic mean in (3)?

**AS:** Good question. We use the harmonic mean here because we assume a row configuration at the calculation of the lateral diffusive mixing fluxes between macropore and matrix as there is a vertical interface between the two domains. We will justify the use of the different means in the revised paper.

**R1:** Page 7, line 24: it is still not clear what are C and DM and their meaning is the opposite of what is defined in the caption of Figure 2.

**AS:** Yes, you are generally right with your criticism on Figure 2 and its caption. Please see our revision of Figure 2 above (Fig. 4). As you can see, we added the equation for the circumference of a macropore grid element which shows that the circumference is a function of the macropore diameter. We will also describe the underlying concept of the pfd in more detail in the methods section.

**R1:** Page 8, Lines 4, 7: what is the depth of the soil samples? Please specify also the initial soil saturation used in the model.

**AS:** We will explain this in our revised paper in more detail. There was a 1 m² plot which was subdivided into 10 depths (every 10 cm vertically) and in each of the depths, there were ten samples taken (every 10 cm horizontally) (in total 100 soil samples).

Furthermore, you can see the sampled depths at the observed mass profile in Figure 3c of our paper. And the initial soil moistures at the three sites are listed in Table 1.

**R1:** Page 9:, eq 7: how was this coefficient computed?

**AS:** Please see Figure 1 and our response to your first major comment. The relation of the flux rate of a macropore or the $k_{pfd}$ with the radius of a macropore was measured by Zehe et al. (2001) at the Spechtacker site. Flow experiments with soil cores containing differently sized macropores were conducted to determine the hydraulic conductivity of macropores with different radii assuming macropore flow is dominating in these soil cores. We will clarify this context within our revised paper.

**R1:** Page 9, section2.4.3: the proposed macropore structure has many degrees of freedom. Is it possible to calibrate / validate such a model with infiltration measurements?

**AS:** Thanks for this comment. As stated above, several of our parameters are observable in the field and in the presented case we were able to derive them from detailed data. If these data are not available, but we still work at sites where anecic worm burrows are the dominant macropore type, we still rely on the regression shown in Figure 1 because its functional form is in line with the law of Hagen-Poiseuille. We would of course remain with the macropore diameter distribution and the depth distribution as unknown, which need to be calibrated on tracer data. This will however be a subject to equifinality (because this is a generic problem), as shown in e.g. Wienhöfer and Zehe (2014). We will better explain this in the discussion of the revised manuscript.

**R1:** Page 9, line 20: Which kind of sensitivity analyses is performed? Sensitivity of which output of the model? From table 2 I understand that the parameters used in the sensitivity

analysis are evenly spaced in the parameter space. Usually in MC approaches the parameters are randomly selected from the parameter space.

**AS:** Good point, we indeed used evenly spaced parameters within the presented range in Table 2. We will therefor rename our sensitivity analyses and leave out the label "MonteCarlo". Generally, to check the sensitivity of our model to various input parameters we used the simulated solute mass profiles and checked if they show explainable behaviours in relation to the input parameters.

**R1:** Page 9, lines 31 – 33: this part should go in the discussion.

**AS:** We think that a short introduction and explanation why we especially chose these three parameters (ks, dmac, nmac) is important for the understanding of the presented sensitivity analyses and thus, we will leave this short passage in the methods section.

**R1:** Page 10, line 1: This sentence is not clear here. I suggest to describe the observations (and the possible difference with the mode outputs) in the methods section. Please provide more information about how these observations are obtained. The real process is three-dimensional. How are these concentrations obtained? Are they an average of the concentrations in different layers?

**AS:** Yes, we think you are right here. We will replace this sentence to the methods section and will provide further details on the underlying field experiments (please, see also our responses above). But yes, bromide concentrations were averaged over each sample depth to compare them with our 1-D results.

**R1:** Page 10, line12: change 'suggest' with 'suggests'

**AS:** Thanks, we will correct that.

**R1:** Page 11 , line 3: please specify which are the three values of ks considered

**AS:** low ks: $1 \cdot 10^{-6}$ m/s; medium ks: $2{,}5 \cdot 10^{-6}$ m/s; high ks: $1 \cdot 10^{-5}$ m/s. Actually, the range of these values is also listed in Table 2 but we will also insert these values into the text.

**R1:** Page 11, Section 3.3.2: please discuss why in figures 7a and 8d the concentration increases between depths -0.15 and -0.4 for all the macropores diameters considered.

**AS:** Thank you very much for pointing this out. A prerequisite for exchange between the pfd and the matrix is saturation in the grid elements of the pfd and this occurs at first in the shallowest macropores. The exchange is then driven by two concurring factors a) the potential gradient which increases with depth, reflecting the decline of the soil water content with depth and b) the harmonic mean of ks and k($\square$) which strongly decrease with

depth. This leads to a tradeoff and likely to a maximum. We will further explore this and show the concurring controls in an appropriate figure of the revised manuscript.

**R1:** Page 12, line 30: This sentence in not correct: to prove this sentence, the sensitivity analysis should be performed by perturbing the parameters of the real-case experiment. However the parameters of the real-case experiment are not among the ranges considered in the sensitivity analysis.

**AS:** Thank you. At the revision of the parameters we have found a mistake in the values of the macropore diameter. We correct it and now the parameters of the study site Spechtacker are indeed within the range used in the sensitivity analyses (Table 1).

Table 1: Value range of saturation conductivity (ks), diameter of macropores (dmac) and number of macorpores (nmac)

| Parameter | Value range |
| --- | --- |
| $k_s$ [m/s] | $10^{-6} - 10^{-5}$ (step: $1 \cdot 10^{-6}$) |
| $dmac$ [m] | $0.0035 - 0.008$ (step: $0.0005$) |
| $nmac$ [-] | $11 - 20$ (step: 1) |

Table 1 shows the revised parameter ranges of the sensitivity analyses. The parameters at the study site Spechtacker are within the same ranges: ks = $2.50 \cdot 10^{-6}$ m/s; dmac = 0,005 m, nmac = 16.

**R1:** Page 12, lines 31-33: this modelling detail should be specified in the model setup. Which is the computational cost of the model when using 2 million particles?

**AS:** Sorry, but the 2 million particles here were a mistake. It should have been 1 million. The simulation of the Spechtacker experiment with 1 million particles runs about five minutes.

**R1:** Page 13, line 18: From figure 5, the sensitivity of the results with respect the considered variation of ks is quite small. What are the differences when changing kpfd?

**AS:** Sorry, but we do not really see that the sensitivity of the model towards different ks values is small. There are significant differences in the solute concentration profiles at the end of simulation (24h) as shown in Figure 5 of our paper. You are right when referring to earlier simulation times. In the first few hours, there is indeed just a slight difference between the different ks values but we also discuss this issue.

Changing $k_{pfd}$ would probably have just a marginal effect because the hydraulic conductivity and velocity of macropores are always so high and the distances small that particles get instantaneously transported to the bottom of the macropores or to the last unsaturated grid element, respectively.

**R1:** Figure 2, caption: the saturated hydraulic conductivity of a macropore here is indicated with ks, while in the main text (page 5, line 8) it is indicated with kpfd. I think the same notation should be used for these two variables. From the figure I am not sure to understand the difference between the diameter of macropore (DM) and the circumference of a grid element (C ). Why the length of a grid element is expressed as dz(z) ?

**AS:** Yes, you are right. Figure 2 of the original paper has some weaknesses. Please see our revised Figure 4 above and its caption and also our previous responses. We hope that the difference between macropore diameter and the circumference are now obvious. Further, we will use a consistent notation for the parameters.

**R1:** Figure 2b: this figure is really not clear and not well explained in the caption. Do the three cylinders correspond to three different time-steps? The times should be better indicated in the figure and the caption should describe what is happening in the three steps.

**AS:** The revised figure also contains a new and better understandable presentation of the macropore filling (Figure 4b). Yes, the three cylinders correspond to three exemplary time steps and we also indicate and explain these time steps in the figure caption.

**R1:** Table 2: please consider using a parameter range that covers the parameters used in the Spechtacker test. Not only it is important to see the sensitivity of the model in this test, but I think the results obtained for the Spechtacker are quite interesting, having a very deep infiltration.

**AS:** Thanks, and as already mentioned above we revised the parameter ranges.

**R1:** References: please check the references: Zehe and Bloshl in not in the correct place. Sometimes there is an 'and' before the last author, sometimes not.

**AS:** Thank you, that is right. We will correct that.

Thank you very much,

Alexander Sternagel on behalf of all authors

**References**

van Schaik, L., Palm, J., Klaus, J., Zehe, E., and Schröder,B.: Linking spatial earthworm distribution to macropore numbers and hydrological effectiveness, Ecohydrology, 7, 401–408, https://doi.org/10.1002/eco.1358, 2014.

Wienhöfer, J., and Zehe, E.: Predicting subsurface stormflow response of a forested hillslope–the role of connected flow paths. Hydrology and Earth System Sciences, 18(1), 121-138, https://doi.org/10.5194/hess-18-121-2014, 2014.

Zehe, E.; Flühler, H.: Preferential transport of isoproturon at a plot scale and a field scale tile-drained site, J. Hydrol. 247 (1-2), 100–115, https://doi.org/10.1016/S0022-1694(01)00370-5, 2001.

Zehe, E. and Jackisch, C.: A Lagrangian model for soil water dynamics during rainfall-driven conditions, Hydrol. Earth Syst. Sci., 20, 3511–3526, https://doi.org/10.5194/hess-20-3511-2016, 2016.

---

## Author Comment (AC2) · 22 May 2019

**Response to Comments of Anonymous Referee #2**

On behalf of all co-authors I sincerely thank the Anonymous Referee #2 for his thoughtful and detailed assessment of our work.

Major Comments

**R2:** As mentioned in the Introduction and in the Conclusions this model is similar to a double-continuum one. In my opinion, it would be interesting if you could elaborate this similitude, ideally linking the parameter of your model with the ones of a classical double approach (with appropriate references)

**AS:** This similarity arises from the fact that both the LAST-Model and double-domain models work with two different domains. But apart from that, the two domains of our model and these of the double-domain models have not much in common because we really established a separate, physically and geometrically described macropore domain with the particle-based Lagrangian approach to simulate water flow and solute transport. The other double-domain models rely on separated overlapping continua and in some of these models water flow is again simulated by the Darcy-Richards equation assuming immobile water fractions or different hydraulic conductivities.

Both, our Lagrangian approach and double-domain models such as HYDRUS 1-D by Šimůnek and van Genuchten (2008) use, however, the same standard parameters like a spatial description of the soil domain with total length and grid element lengths, simulation time and time stepping, initial soil moisture and also soil hydraulic properties.

As proposed by you and the other reviewers, we now additionally compared LAST with HYDRUS 1-D using the same three infiltration experiments. As you can see in Figure 1, at the well-mixed study sites 23 and 31 HYDRUS 1-D performs well in accordance to the observed values and its model results are similar to our simulation results with just slight deviations but which are in the range of uncertainty. In contrast, at the preferential flow site Spechtacker HYDRUS 1-D with its double-domain approach is not able to simulate well the highly heterogeneous, observed solute mass profile. Here, our model performs much better in comparison. We will discuss these results in our revised paper in more detail.

[Figure]

Figure 1: Solute mass profiles at our three study sites simulated with HYDRUS 1-D (lower part) and compared to the mass profiles simulated with our LAST-Model (upper part)

**R2:** Pag.4, line 23: here I am a bit confused about how do you compute the diffusive mixing: it is not the entire solute mass in a grid element given by the mass of all the present water particles?

**AS:** Absolutely true and sorry for the confusion. We will revise this part. Generally, diffusive mixing among all particles is calculated after each displacement step by summing up the entire solute mass in a grid element and dividing it by the amount of all present water particles. In this way, each particle gets a new solute mass in every time step.

**R2:** Pag.4, line 26: do you have some criteria to define a "sufficiently fine" grid?

**AS:** Generally, the grid elements of both matrix and pfd are necessary to create small spatial discretizations for the calculation of the new state variables (soil moisture, solute concentration, hydraulic conductivity) in each time step and in this way to register even slight spatial and temporal alterations of the state variables So, the grid elements have to be fine enough to ensure this proper registration of changes of the state variables and to ensure that the system remains stable without oscillations during simulation but also not too fine so that simulation times increase exorbitantly. Please also see the study of Zehe and Jackisch (2016) who determined the influences and sensitivities of different grid element sizes to the particle-based Lagrangian approach.

**R2:** Pag.5 line 10: could you please list all the parameters of the model and do not only refer to Figure 2 in order to better clarify how many parameters the model counts? Is the pfd characterized by the 17 parameters given in the caption of Figure 2? Is the macropore diameter "dmac" (in the text) equal to "D_M" (in Figure 2 and in Section 2.3.i)? In Table 1 we have 16 parameter for the soil description (7 for the soil type and 9 for the macropores domain) + 8 for the experiment conditions + 4 for the numerical implementation.

**AS:** We apologize for Figure 2 in our original paper which is indeed hard to understand. We already revised the figure and its caption, please see Figure 2 below. In general, the pfd is characterized by 9 parameters (the macropore lengths, diameter, distribution factors, grid element length). The other characteristics like the volume, lateral area etc. depend on these 9 parameters and the flow rate depends on the macropore diameter (compare Fig. 3 of this response below).

In our revised paper we will better describe how we obtained observable parameters and how we calculated or derived other parameters from those observables. And yes, dmac is equal to $D_M$, we will clarify this notation.

[Figure]

Figure 2 (i.e. Figure 2 of the revised paper): Conceptual visualization of a) macropore structure and cubic packing of particles within the rectangle of a cut open and laid-flat grid element cylinder, b) macropore filling with gradual saturation of grid elements, exemplarily shown for three time steps ($t_1$-$t_3$) whereby in each time step new particles (differently coloured related to the current time step) infiltrate the macropore and travel into the deepest unsaturated grid element c) macropore depth distribution and diffusive mixing from macropores into matrix.

[Figure]

Figure 3: Linear Regression of the flux rate within the macropore on the macropore radius at the study site Spechtacker (Zehe et al. (2001)). This relation was derived from measurements of saturated flow through undisturbed soil columns containing worm burrows.

**R2:** Pag. 6, line 6: how is the matrix potential gradient between the first two grid elements computed?

**AS:** It is calculated as the difference of psi between the first two grid elements in each time step at the beginning of the infiltration routine. Please note, that the first grid element belongs to the soil surface (z = 0) and the second actually to the first grid element right beneath the soil surface (z = 5 cm). We will revise the explanation of this calculation.

**R2:** Pag. 7, line 11: why does your model "generally" divide the total amount of macropores into 3 parts? Could you please explain the meaning and the effect distribution factor?

**AS:** We assume that a macropore distribution with three different depths is a sufficient approximation of the observed macropore depth distribution at the study site Spechtacker. Nevertheless, as a variable macropore depth distribution might be observed at other sites we think that the model needs to be more flexible in this respect.

The distribution factors distribute the total amount of macropores among the three defined depths and determine in this way in which depths and to which extent water and solute masses are diffusively exchanged between the macropore and the matrix. This distribution is based on real-observed data of the macropore network at the Spechtacker site. As already mentioned, we will implement an additional section/paragraph to properly explain the model database and the derivation of all the parameters.

**R2:** Pag. 7, line 20: why do you use the harmonic mean to compute a sort of "effective" hydraulic conductivity? Usually, the effective hydraulic conductivity for heterogeneous media in parallel configuration is computed as the arithmetic mean of the 2 conductivities…is your model sensitive to this choice?

**AS:** Yes, you are of course right for parallel configurations. But here we use the harmonic mean because we assume a row configuration at the calculation of the lateral diffusive mixing fluxes between macropore and matrix as there is a vertical interface between the two domains. We will add this explanation to the revised paper.

**R2:** Eq(7): is the number "2884.2" result of a calibration? Could you please provide some details about it? Have you calibrate some parameter of your model to fit the experimental data? Could you please provide some details about how to use the model to interpret experimental data?

**AS:** The relation of the macropore flux rate or the $k_{pfd}$ to the radius of a macropore was measured by Zehe and Flühler (2001) at the Spechtacker site. This relation was derived by measurements of saturated flow through undisturbed soil columns which were centered around worm burrows with the assumption that flow through these macropores dominated. When normalizing the measured flow with the cross sectional area of the macropores, they obtained a linear dependence of the average flow rate with the macropore radius which is in

line with Hagen-Poiseuille. Please see Figure 3 of this response which shows the linear regression to determine this dependence. We will explain this relation in our revised paper in more detail.

**R2:** Pag.9, line 21: in my opinion, the way you describe your sensitivity analysis is a bit vague.. how do you conclude that ks, dmac (=Dm?), nmac are the most sensitive parameters? How do you conclude that ks is "probably the most sensitive parameter" (Pag.9, line 31) ?

**AS:** Sorry, that our description is unclear. We will explain our sensitivity analyses more properly in the revised paper. In general, due to the model structure we early assumed that it would be logical if these parameters were most sensitive because dmac and nmac mainly define the new macropore domain and ks plays a crucial role in the infiltration process, the particle displacement within matrix and even the macropore-matrix diffusion.

**R2:** Pag. 10: Result section: I am sorry, but for me it is not clear how do you select the parameters of your model to simulate the tracer mass in the respective depths, could you please state more clearly which observables you had, which parameters you compute from measurements etc.

**AS:** We agree that this needs to be better explained. We will implement an additional chapter to properly explain the model database and the derivation of all the parameters. Due to the extensive mapping of the macropore network at our study site Spechtacker, we had a detailed database containing information on macropore numbers, depths and diameter distributions. From these data it was able to derive our nmac, dmac, depth distribution and the distribution factors. Please see the following Figure 4 to get an idea of these observations.

[Figure]

Figure 4: Patterns of dye tracer (a+d) and worm burrows as well as the measurement of distribution, lengths and diameters of those macropores in different horizontal layers (d) at the study site Spechtacker (taken from van Schaik et al. (2014)).

**R2:** Pag. 10, line 23: do you have an explanation about the greatest difference in profiles between 0.15 and 0.35m depth?

**AS:** Good question. We think that simulation in this region is difficult a) because of the lateral network of endogeic worm burrows which are completely unknown and not represented in the model and b) due to the influence of the nearby plow horizon in 30 - 35 cm depth. We will stress that soil properties are uncertain in this region.

**R2:** Pag. 10, line 32: here, as in Figure 9, it looks that the results are sensitively depending on the Configuration (1,2,3) … there is a way to parametrize the different configurations in order to study and quantify the sensitivity of the model to the different configurations?

**AS:** Yes, there is again the problem with the imprecision of our description of the database and model parametrization. As already mentioned, we will implement an additional section/paragraph to address this issue and to explain how the different configurations were parametrized.

**R2:** Pag.12, line 28: here, you conclude that macropore-matrix exchange should be modelled deriving an "effective conductance", even if it is the first time you introduce this term. I suppose you refer to the coefficient in Eq.6, but I would specify it.

**AS:** Yes, you are right. We indeed refer to the diffusive mixing flux calculated by equation 6. We think we will introduce this term at the presentation of equation 6 in the methods section.

**R2:** Pag. 13, line 3: I agree that "further field experiments on a variety of differently structured soil is necessary", however, from my point of view, it is not clear how do you parametrize these differently structured soils as well as do you parametrize the spatial heterogeneity of the macropores network (Pag. 13, line 35)

**AS:** In general, our model is able to consider also several, differently structured soil layers with different soil parameters and not just one homogeneous soil type. As stated above, several of our parameters are observable in the field and in the presented case we were able to derive them from detailed data. If these data are not available, but we still work at sites where anecic worm burrows are the dominant macropore type, we still rely on the regression shown in Figure 3 because its functional form is in line with the law of Hagen-Poiseuille. We would of course remain with the macropore diameter distribution and the depth distribution as unknown, which need to be calibrated on tracer data. This will however be a subject to equifinality (because this is a generic problem), as shown in e.g. Wienhöfer and Zehe (2014). We will better explain this in the discussion of the revised manuscript.

**R2:** Pag. 13, line 13: in my opinion it is not so straightforward how do you transfer the concept of cubic particle storage and hydraulic radius to any kind of macropore geometry.

**AS:** Good point, we agree that these concepts have certain limitations especially for complex geometries. This is a generic problem with respect to frictional loss and exchange. Such complex geometries could be expected when dealing with soil cracks but when referring to biologically generated macropores like worm burrows, degraded plant roots or even for example ant channels, we think that the resulting macropore geometries would be simple enough to apply the concept of cubic packing and hydraulic radius.

Minor Comments

**R2:** Eq. (1): please check this equation. I suppose a "+ z_i(t)" after the equal is missing and the format is different from the other equation in the manuscript.

**AS:** Yes, you are absolutely right. Thank you. We will correct that.

**R2:** Pag. 4 line 5: could you provide some details about the soil water retention curve used to compute the diffusivity from the hydraulic conductivity?

**AS:** It is a typical soil water retention curve with the relation $\frac{\partial \psi}{\partial \theta}$. Multiplied with the hydraulic conductivity you can obtain the diffusivity. Thus, in each time step in the particle displacement routine we compute this relation with the current values for psi and theta to obtain the diffusivity for each particle.

**R2:** Pag. 4, line 6: I guess that Z is a random uniformly distributed number "between 0 and 1";

**AS:** Actually, it is between -1 and 1. But thanks for calling our attention. We will clarify this in the text. With this range the particles are allowed to move vertically up- and downward.

**R2:** Pag.4, line 21: could you please provide some details about the numerical implementation of the model (e.g. programming language etc)?

**AS:** Yes, you are right. This is something that is still missing in our paper. We will add some information on this issue. The programming language is MATLAB and the model runs were performed on a casual personal computer with moderate computational power (e.g. Intel i3, 4 GB RAM).

**R2:** Pag. 6, line 4: typo: please write consistently k_m1 with Eq.(3) as well as n_mac introduced in Pag.5, line 6.

**AS:** Sorry, yes. We will revise the text for a consistent notation.

**R2:** Pag 6, line 8: is the simulation time step "dt" or "Delta t"?

**AS:** Again, we will look for a consistent notation. Simulation time step will be "Delta t".

**R2:** Pag 7, line 22: please correct a typo: "matric" potential

**AS:** Thank you, we will correct that.

**R2:** Pag. 12, line 33: Here you say that you need at least two million particles, but I suppose the minimum number of particles you need is proportional to the observation area, isn't it?

**AS:** We are sorry, because there is a mistake. We have to correct the particle number to 1 million.

Further, we think that the total amount of particles does not necessarily depend on the domain extent. We think that it instead depends on the total amount of water stored within the domain. Particles must have a sufficient volume and mass, and to scale these measurements you can adjust the total particle number, e.g. at high water masses within a domain you have to select a higher particle number to avoid that a single particle carries an immense water mass and consequently has also a too large volume. This case can also arise in small domains.

But yes, when simulating a hillslope you generally need more particles because the large spatial extent of the hillslope usually implies also a high number of stored water masses.

**R2:** Pag. 13 line 38: you conclude that your model provides high computational efficiency with short simulation times, could you please provide further details?

**AS:** The simulation of the infiltration experiment at the study site Spechtacker with the selected parametrization runs for about 5-10 minutes on a casual personal computer with moderate computing power. Without an active pfd (e.g. at the other two infiltration tests) the model runs even faster (couple of minutes). When performing these simulations on a high performance computer or work station, you probably could also run several model simulations in parallel within minutes.

And further as mentioned in the introduction of our paper, the comparable echoRD model of Jackisch and Zehe (2018) has simulation times 10 -200 longer than real time.

We will expand our discussion to provide this information in the revised paper.

Thank you very much,

Alexander Sternagel on behalf of all authors

**References**

Jackisch, C., Zehe, E.: Ecohydrological particle model based on representative domains, Hydrol. Earth Syst. Sci. 22 (7), 3639–3662, doi:10.5194/hess-22-3639-2018, 2018.

Šimůnek, J., van Genuchten, M. T.: Modeling nonequilibrium flow and transport processes using HYDRUS, Vadose Zone Journal 7 (2), 782–797, doi:10.2136/vzj2007.0074, 2008.

van Schaik, L., Palm, J., Klaus, J., Zehe, E., and Schröder,B.: Linking spatial earthworm distribution to macropore numbers and hydrological effectiveness, Ecohydrology, 7, 401–408, https://doi.org/10.1002/eco.1358, 2014.

Wienhöfer, J., and Zehe, E.: Predicting subsurface stormflow response of a forested hillslope–the role of connected flow paths. Hydrology and Earth System Sciences, 18(1), 121-138, https://doi.org/10.5194/hess-18-121-2014, 2014.

Zehe, E.; Flühler, H.: Preferential transport of isoproturon at a plot scale and a field scale tile-drained site, J. Hydrol. 247 (1-2), 100–115, https://doi.org/10.1016/S0022-1694(01)00370-5, 2001.

Zehe, E. and Jackisch, C.: A Lagrangian model for soil water dynamics during rainfall-driven conditions, Hydrol. Earth Syst. Sci., 20, 3511–3526, https://doi.org/10.5194/hess-20-3511-2016, 2016.

---

## Author Comment (AC3) · 22 May 2019

**Response to Comments of Anonymous Referee #3**

On behalf of all co-authors I sincerely thank the Anonymous Referee #3 for his thoughtful and detailed assessment of our work.

Major Comments

**R3:** The modelling strategy have been proposed to overcome challenges related to dual domain models, however, there no quantitative comparison between the dual domain methods proposed in Seven and Germann, (1981) or Nezhad et all (2010), and the model proposed by authors in this manuscript. A further analyses is required to compare the results achieved from the extended work and the original LAST model as well as results that can be achieved via dual domain theory. These quantitative comparisons are required, particularly, for clarification of discussions in lines 25-30 if the page 12.

**AS:** We thank the reviewer for this comment. The main objective of our study is to propose an alternative approach to model the interplay of water flow and solute transport in structured heterogeneous soils containing macropores using a full Lagrangian approach. With their study, Zehe and Jackisch (2016) have already successfully tested this particle-based Lagrangian approach with the linear mixing assumption against a 1-D Richards solver.

Further, in the revised paper version we will additionally test the solute transport routine of our model with HYDRUS 1-D. To this end, please see Figure 1 of this response which shows the results of the simulation of our three infiltration tests with HYDRUS 1-D compared to the results of our LAST-Model.

As you can see, at the well-mixed study sites 23 and 31 HYDRUS 1-D performs well in accordance to the observed values and it is also similar to our simulation results with just slight deviations but which are in the range of uncertainty. In contrast, at the preferential flow site Spechtacker HYDRUS 1-D with its double-domain approach is not able to simulate well the highly heterogeneous, observed solute mass profile. Here, our model performs much better in comparison. We will discuss these results in our revised paper in more detail.

[Figure]

Figure 1: Solute mass profiles at our three study sites simulated with HYDRUS 1-D (lower part) and compared to the mass profiles simulated with our LAST-Model (upper part)

**R3:** Some new parameters have been introduced in the new model, which may not be physically measurable such as dimension of the micropores and considering the authors effort for simulation of field data, it has not been proposed/specified how values of these parameters can be identified.

**AS:** We thank the reviewer for this comment. Several parameters of the pfd like the number of macropores, their diameter and dephts are directly measurable in the field. We will better explain this in an additional section/paragraph in the revised manuscript and clarify how we obtained our model parameters from these observables, e.g. also with further figures (see Figure 2 below). With this Figure 2, we can explain that the dimensions of macropores (depth, diameter) are indeed physically measurable in field experiments. As you can see, horizontal soil profiles were excavated in different depths and the number of macropores, their lengths and diameters were measured. From this dataset we derived the parameters of the pfd with dmac, nmac, macropore depths and also the distribution factors. Note that also the flow rate in macropores is based on measurements of saturated flow through undisturbed soil columns, which were centered around worm burrows. These measurements revealed a clear linear dependence of the flow rate on the macropore radius, which is in line with Hagen-Poiseuille's law (Figure 3).

[Figure]

[Figure]

Figure 2: Patterns of dye tracer (a+d) and worm burrows as well as the measurement of distribution, lengths and diameters of those macropores in different horizontal layers (d) at the study site Spechtacker (taken from van Schaik et al. (2014)).

$q_m = 2884.2\ r^2$
$R^2 = 0.803$

Figure 3: Linear Regression of the flux rate within the macropore on the macropore radius at the study site Spechtacker (Zehe et al. (2001)). This relation was derived from measurements of saturated flow through undisturbed soil columns containing worm burrows.

**R3:** Discussion regarding computational efficiency of the proposed model has not been presented sufficiently, and for example in page 13 line 39 duration of simulation has been presented without identifying which machine have been used and also duration of simulation with other possible model have not been compared. With our such complete comparisons, discussions on efficiency of the method would not add any scientific knowledge to the readers.

**AS:** Yes, you are right. We will add some more information about the computational setup and efficiency. We used the programming language MATLAB on a casual personal computer with moderate computational power (e.g. Intel i3, 4 GB RAM). Further, we compared our model efficiency at least against one other model, the echoRD model (see page 3, line 8) which has simulation times up to 10 – 200 longer than real time.

The simulations of the first two well-mixed cases without considering an active pfd run even faster in a couple of minutes. When performing these simulations on a high performance

computer or work station, you probably could also run several model simulations in parallel within minutes. Further, the amount of total particles has a major impact on the computational efficiency: A double amount of particles results in a more than double increase of the simulation time.

**R3:** Some of the results presented in the paper are obvious and do not need complex modelling methods to be implemented. For example discussions presented in page 12 lines 15-20, can be achieved using other methods and perhaps developing proposed model was not required to understand these. Perhaps if authors compare their results with other results achieved using other methods which capture the effects of macropores, more valuable finding will be presented. Authors should make the results section more focused on the capacity of new strategy used for modelling micropores and their interactions with soil matrix.

**AS:** We agree with the reviewer that some results of the sensitivity analyses are straightforward. Nevertheless, we think their presentation is necessary to allow the reader to check if our Lagrangian approach with the macropore domain reproduces these results as the model concept is new and the exchange between both domains does not rely on an extra parameter like a leakage coefficient, e.g. used in dual models (Gerke, 2006).

We agree that the ability of our LAST-Model to reproduce the fingerprint of macropore flow observed in the tracer profile at the Spechtacker site is the main part of the results section and we will put more emphasis on this. In this respect, we are not aware of many other model studies which reproduce preferential flow fingerprints using a model structure relying on observed data. We think that the comparison with HYDRUS 1-D corroborates the feasibility of the model.

Minor comments

**R3:** Simulation domains have not been explained sufficiently in the text, and mainly some figures have been presented which are not enough to understand the problem being simulated.

**AS:** Yes, you are right. Your criticism is in line with the other reviews. We will add further information on the simulation domains in the text and also revise the Figure 2 of our paper and its caption, e.g. with this revised Figure 4 and caption:

[Figure]

Figure 4 (i.e. Figure 2 of the revised paper): Conceptual visualization of a) macropore structure and cubic packing of particles within the rectangle of a cut open and laid-flat grid element cylinder, b) macropore filling with gradual saturation of grid elements, exemplarily shown for three time steps ($t_1$-$t_3$) whereby in each time step new particles (differently coloured related to the current time step) infiltrate the macropore and travel into the deepest unsaturated grid element c) macropore depth distribution and diffusive mixing from macropores into matrix.

We think the revised figure is now easier to understand. The explanation of all pfd parameters was moved from the caption to the text.

**R3:** A complete description of boundary conditions and initial conditions for simulation domains are required.

**AS:** We will add more information on the boundary conditions. At the upper boundary we have a variable flux boundary describing infiltration of precipitation water into the soil with a Darcy flux and at the lower boundary we assume no-flux conditions.

The initial soil moisture of the matrix is listed in Table 1 of our paper. Further, there is no solute initially stored within the soil and the macropres as well as the surface storage are also completely empty at simulation begin. We will add more information on the boundary conditions in the revised paper.

**R3:** discussion on time step in page 6 lines 20-25 is vague and needs to be clarified. It will be helpful that author visualise the discussion and king it more understandable.

**AS:** Yes, we have to revise the section about the time stepping and macropore filling. Generally, our model can work with variable time stepping as it is not subject to numerically stability criteria. In fact, we select the time step such that the particle displacement per time step equals the maximum depth of the pfd and subsequently we shift excess particles to the deepest unsaturated grid element. In this way we gradually fill the macropores from the bottom to the top (see Fig. 4b of this response above).

**R3:** )If I understood correctly LAST model is the same as the model developed by Zehe and Jackisch (2016). I suggested that author call it as their model or the model developed by

Zehe and Jackisch (2016), i.e., rewrite lines 9-11, I suggest "Our LAST-Model (Lagrangian Soil Water and Solute Transport) developed by Zehe and Jackisch (2016) relies on the movement of water particles carrying a solute mass through the soil matrix and macropores. We advance this model by two main extensions: a)..."

**AS:** Sorry, if there is a misunderstanding. We try to make it clearer in the revised paper. Zehe and Jackisch (2016) just developed the basic idea of using a particle-based Lagrangian approach to simulate water flow in well-mixed soil domains. Now, with this study we extended this basic model by solute transport and a macropore domain and also developed the name of this new model: "LAST-Model". As this name already suggests, it is mainly about solute transport and therewith essentially different to the original model of Zehe and Jackisch (2016) only treating water flow.

Thank you very much,

Alexander Sternagel on behalf of all authors

**References**

Gerke, H.H.: Preferential flow descriptions for structured soils, J.PlantNutr.SoilSci., 169, 382–400, https://doi.org/10.1002/jpln.200521955, 2006.

van Schaik, L., Palm, J., Klaus, J., Zehe, E., and Schröder,B.: Linking spatial earthworm distribution to macropore numbers and hydrological effectiveness, Ecohydrology, 7, 401–408, https://doi.org/10.1002/eco.1358, 2014.

Zehe, E.; Flühler, H.: Preferential transport of isoproturon at a plot scale and a field scale tile-drained site, J. Hydrol. 247 (1-2), 100–115, https://doi.org/10.1016/S0022-1694(01)00370-5, 2001.

Zehe, E. and Jackisch, C.: A Lagrangian model for soil water dynamics during rainfall-driven conditions, Hydrol. Earth Syst. Sci., 20, 3511–3526, https://doi.org/10.5194/hess-20-3511-2016, 2016.

---

## Author Comment (AC4) · 27 May 2019

**Response to Comments of Michael W. I. Schmidt, Sandra Werthmüller and Jasmin Kesselring**

On behalf of all co-authors I sincerely thank Prof. Schmidt and his students for their thoughtful and detailed assessment of our work. We appreciate the idea that students work on reviews of scientific papers and contribute to the discussion process. We think that it is a great opportunity for them to get an idea of the scientific publishing process and insights into the work of a researcher.

General Comments

**R4:** In general, we think that the manuscript has a good structure and one can follow the development of the model the way it is described in the paper. However, we think that the introduction is slightly too long compared to the rest of the manuscript.

**AS:** Thank you for your general positive assessment of our work. We think that after the revision of our paper with the addition of further text passages and figures, the relation between introduction and the other chapters will be better balanced.

**R4:** For us as beginners in the field, it is hard to understand why your model is innovative. Could you explain at the beginning of the paper what makes your model innovative compared to others in the field? And how your work is embedded in the broader work of soil water modelling? We understand that the paper is about discussing the development of a new model and is thus theoretical. However, we think a more practical description of the use of the model would be nice. For instance: For which studies is this model a must have addition? We also think that the model would have to be compared to more than one practical study to fully be called a valid model. […]

**AS:** Sorry, if this was not clear to you. We will revise the introduction. In short, commonly used hydrological models use the Darcy-Richards equation to simulate subsurface water flow. Many studies have shown the validity of this approach under well-mixed conditions in homogeneous soils. But also several studies have proven that the Darcy-Richards approach frequently fails when it comes to preferential flow through macropores in heterogeneous soils and due to rainfall-driven flow conditions. To overcome this weakness we propose our alternative particle-based Lagrangian approach. The differences are that we represent water masses as distinct particles and we are able to follow and describe the trajectory of each single particle through the system. We think that this until now only rarely applied approach is very promising to address the preferential flow issue and also the associated solute transport. With our study we want to evaluate and prove the validity of particle-based Lagrangian models.

And yes, you are right with your suggestion that our work is theoretical. As we are still just at the beginning of the development of our model it is difficult to describe its practical use in the future. When further adding a reactive transport routine and extending the model to 2-D it could be a practical tool to assess the risk of pesticide leaching on field sites or even on entire hillslopes.

Moreover, we consider to perform another simulation of an infiltration test and also to compare our model against the commonly used hydrological model HYDRUS 1-D in the revised version of our paper.

To this end, please see Figure 1 of this response below which shows the results of the simulation of our three infiltration tests with HYDRUS 1-D compared to the results of our LAST-Model.

As you can see, at the well-mixed study sites 23 and 31 HYDRUS 1-D performs well in accordance to the observed values and it is also similar to our simulation results with just slight deviations but which are in the range of uncertainty. In contrast, at the preferential flow site Spechtacker HYDRUS 1-D with its double-domain approach is not able to simulate well the highly heterogeneous, observed solute mass profile. Here, our model performs much better in comparison. We will discuss these results in our revised paper in more detail.

[Figure]

Figure 1: Solute mass profiles at out three study sites simulated with HYDRUS 1-D (lower part) and compared to the mass profiles simulated with our LAST-Model (upper part)

**R4:** Page 3 Line 36 ff: How is the number of bins i and the subdivision into N bins defined? What exactly is the difference between those two and how do you choose the 'perfect' number of bins?

**AS:** Sorry, this is indeed a bit confusing. We will revise that. N is the total amount of bins and can be predefined. Please see also the study of Zehe and Jackisch (2016), who tested how the

number of bins influences the model results. In our model, we use 800 bins. And in contrast, i is the number of the current bin (between 0 - 800) within the displacement routine.

**R4:** Page 4 Line 30-33: Here, you list four subchapters that will follow in the next paragraph. Why not name the actual subchapters according to this list?

**AS:** Yes, you are right. We will adjust the list of the four subchapters.

**R4:** Page 9 Line 31ff: You already start the interpretation of results, why not in the dedicated section (discussion and conclusion)?

**AS:** Yes, sometimes we already started discussing some results in the results section. We did that, because the discussion and conclusion of these results are obvious and logical. Thus, we shortly mention them within the results section and do not come back to them in the discussion. In the discussion, we concisely refer to the main objectives of our study mentioned in the introduction.

**R4:** The layout of your references makes it hard to differentiate references. We also noticed that a lot of citations and references you used are from the same authors. We were wondering, if there are other scientists that are working on the same problem to which you could compare your results with.

**AS:** Indeed, a differentiation of the references is difficult. We consider to revise the layout. Further, there are not many studies and researchers dealing with the still relatively new particles-based approach and we think that we referenced all the crucial studies related to our topic.

Detailed comments

**R4:** The abbreviation for confer is cf.  not c.f.  It is used inconsistently in the manuscript

**AS:** Yes, thanks. We will correct that.

**R4:** Page 1 Line 34: become a major issue (change an to a)

**AS:** Thanks, we will correct that.

**R4:** Page 4 Line 24ff:  This sentence is a bit difficult to understand. Maybe make two sentences e.g. ...corresponding to the molecular diffusion coefficient. Additionally, this needs to be smaller than...

**AS:** Ok, we will consider to revise this passage.

**R4:** Page 6 Line 4: k_m1 or k_m1 with a subscript 1 as in the formula above?

**AS:** Thanks, it should be $k\_m_1$. We will use a consistent notation.

**R4:** Page 8: Has unnecessary empty space

**AS:** Thanks, you are right. We will revise the layout.

**R4:** Page 9 Line 21ff: In this sentence you suggest that the parameter hydraulic conductivity of the matrix ks, diameter of macropores dmac and the amount of macropores nmac are the most sensitive for the model behaviour and simulation results. Please elaborate why and give a reference for it.

**AS:** Sorry, that our description is unclear. We will explain our sensitivity analyses more properly in the revised paper. In general, due to the model structure we early assumed that it would be logical if these parameters are most sensitive because dmac and nmac mainly define the new macropore domain and ks plays a crucial role in the infiltration process, the particle displacement within matrix and even in the macropore-matrix diffusion.

**R4:** Page 9 Line 24ff: In this paragraph you mentioned different configurations for depth distribution and distribution factors. They have the same numbers, which is confusing and makes the text hard to understand. If possible, clarify the difference between depth distribution and distribution factors.

**AS:** Yes, we indeed used the same numbers for two different distributions. We will revise this issue and change the notation, e.g. macropore depth distribution with configurations 1-3 and distribution factors with configurations a-d.

**R4:** Page 13 Line 37 ff: You mention that your model is highly computational efficient and with a short simulations time (about five minutes). How does this short simulation time compare to other similar models? Could you give a reference time? And could you explain how this new model increased computational efficiency?

**AS:** The simulation of the infiltration experiment at the study site Spechtacker with the selected parametrization runs for about 5-10 minutes on a casual personal computer with moderate computing power (e.g. Intel i3, 4 GB RAM). Without an active pfd (e.g. at the other two infiltration tests) the model runs even faster (couple of minutes). When performing these simulations on a high performance computer or work station, you probably could also run several model simulations parallel within minutes.

And further, as mentioned in the introduction of our paper, the comparable echoRD model of Jackisch and Zehe (2018) has simulation times 10 -200 longer than real time.

The reason for the computational efficiency of our model is the fact that we tried to keep the model structure as simple as possible using a combination of appropriate assumptions and basic physical rules.

**R4:** Figure 1: Why are pore size and soil water content equal to each other? (x-axis) Maybe mention in the figure caption how the bin width is calculated.

**AS:** Good question. Related to the velocity or hydraulic conductivity of the matrix (y-axis) the water content and pore size can be seen as equal because big pores contain more water and also the binding forces in these big pores are reduced. Both facts lead to a higher flow velocity. The calculation of the bin width is explained in the text but we will consider to also mention it in the figure caption.

**R4:** Figure 2: In line 3 of the caption: describe DM, LM, dz separately like the other parameters and not as a group. We do not understand what figure b) means. What do the different colours stand for? Describe it better in the text where you reference it as well as in the figure caption

**AS:** Thank you. Your criticism on Figure 2 of our paper is adequate and in line with the other referee comments. We will add a revised version of Figure 2 and a better explanation to the revised manuscript. Figure 2 of this manuscript gives an idea of the revised figure. We will move the definitions of the parameters of Figure 2 to the text.

[Figure]

Figure 2: Conceptual visualization of a) macropore structure and cubic packing of particles within the rectangle of a cut open and laid-flat grid element cylinder, b) macropore filling with gradual saturation of grid elements, exemplary shown for three time steps ($t_1$-$t_3$) whereby in each time step new particles (differently coloured related to the current time step) infiltrate the macropore and travel into the deepest unsaturated grid element c) macropore depth distribution and diffusive mixing from macropores into matrix.

**R4:** Figure 3/4: Is the coloured in area the uncertainty range? Are these different parameters in figures 3 and 4 or why do they have different colours? For us the graphics are also a bit small which makes it difficult to read them. It would be better if the graphics were a bit bigger.

**AS:** Sorry, if this is unclear. Figure 3 of our discussion paper shows the simulated mass profile at the three study sites compared to the obtained data of the real infiltration tests. The rose area shows the model uncertainty/ -changes to different model setups. And Figure 4 of our

discussion paper is part of the sensitivity analyses and the blue area shows the range of different model results dependent on different ks values. Thus, as both figures relate to different issues (Figure 3: re-simulation of real infiltration test, Figure 4: sensitivity analyses with different ks values), we used different colours to emphasize the difference.

**R4:** Figure 9: In all four plots use the same colour for the same configuration number. This makes it easier to see the influence of the different factors on the configurations.

**AS:** Sorry, if there is a misunderstanding. We deliberately used different colours in Figure 9a+b) and 9c+d) as they relate to two different configuration setups (Figure 9a+b): distribution of macropore depths with three different configurations 1-3; Figure 9c+d): Four different configurations 1-4 of distribution factors). We will revise Figure 9 of our paper and the explanation of the different configurations in the text. Please see also our response to your previous comments.

Thank you very much,

Alexander Sternagel on behalf of all authors

**References**

Jackisch, C., Zehe, E.: Ecohydrological particle model based on representative domains, Hydrol. Earth Syst. Sci. 22 (7), 3639–3662, doi:10.5194/hess-22-3639-2018, 2018.

Zehe, E. and Jackisch, C.: A Lagrangian model for soil water dynamics during rainfall-driven conditions, Hydrol. Earth Syst. Sci., 20, 3511–3526, https://doi.org/10.5194/hess-20-3511-2016, 2016.

---

## Author Response (AR2)

**Dear Editor,**

We again thank you for your effort and the coordination of the review process. Your comments as well as the comments of the reviewer were very constructive and helped to significantly improve our manuscript. In line with our response to the reviewer, we considerably revised our manuscript as outlined below:

- We added a passage in the discussions section to specify the possibilities of parametrization of our model, in line with our response to the first comment of the reviewer.
- We revised some other sections and corrected several typos, considering the minor comments of the reviewer.
- Moreover, we again revised our abstract to make it more significant.

Furthermore, our authors' response contains:

- a point-by-point response to the review
- a marked-up manuscript version (all changes made or contents added are highlighted in yellow colour)

Thank you very much,

Best regards

Alexander Sternagel on behalf of all authors

**Point-by-point response to the review**

**Response to Comments of Anonymous Referee #2**

On behalf of all co-authors I sincerely thank the Anonymous Referee #2 for his thoughtful and detailed assessment of our work.

**R2:** In my opinion, the authors replied quite satisfactorily to my first review. They added an additional infiltration test to illustrate the feasibility of the LAST-Model, they compared the results of the model with the ones of a numerical one (HYDRUS-1D) and they (partially) improved the explanation of LAST-model parametrisation. However, there are some typos and I still have some doubts about the feasibility of designing a physically based structure of the preferential flow domain suitable for the model proposed. Specifically, in Section 3.2 devoted to the parametrization of the model, is written: "Horizontal layers in different depths of the vertical soil profiles were excavated and in each layer the amount of present macropores counted as well as the diameters and depths measured". Is this the procedure one should use in order to apply your model? In order to use LAST-model, should one excavate at different depths, until an (unknown) "predefined length" (=LM), count the number of macropores (=nmac), measure their diameters (=dmac) and use these numbers as input parameters for your model? Especially because the model looks extremely sensitive to these parameters.

**AS**: We thank you for the positive comment on our previous revision and for your questions. The answer is yes and no. We indeed recommend mapping of the macropore system if you want to characterize the transport behavior of a field site using tracer experiments. These experiments are very laborious anyway (irrigation application, profile excavation, soil sampling on a grid, tracer extraction etc.), particularly if soil hydraulic properties are further characterized by soil sampling and subsequent multistep outflow experiments. But compared to this general effort, the mapping of a macropore network is relatively easy to realize by excavating one horizontal layer with a size of 1 m by 1 m, simple counting of macropore numbers, measurement of their radii and depths, e.g. by pushing a wire into the macropores. Altogether this procedure takes about 2 h per replicate. Please note that we added this section to corroborate that the geometric properties of the macropore domain are observable and not random factors as criticized in a review of the previous manuscript. Beside the parametrization with experimental data, it is also possible to setup our model by using pedotransfer functions for the soil hydraulic properties and to vary the parameters of the macropore domain by inverse modelling, which needs prior knowledge of the depth of typical macropore systems (e.g. worm burrow networks) and literature data to parametrize macropore flow velocities. This method would reduce time and the amount of work but it could result in equifinality as shown by Klaus and Zehe (2010) or Wienhöfer and Zehe (2014). We will better explain this in the revised manuscript.

**R2:** In the 5.2 Result and Discussion Section is written that a possible explanation for the discrepancy between the model and the data observed at Site 33 is that "examination of the

macropore network were performed on different dates" ... Does it mean that even if I carefully parametrize the pfd zone (with LM, nmac, dmac) I have to consider a temporal evolution of this parameters, depending e.g. to saturation conditions? It looks like a long and expensive preliminary work, for a relatively simple 1D double domain water flow and tracer transport model or am I missing something? I write "relatively simple" in the sense that there are many hypothesis underlying your model (e.g. no lateral exchange, diffusive exchange between solutes in the matrix, reverse diffusion from the matrix into the macropores...).

**AS:** As stated above, performing tracer field experiments is always time and work expensive anyway. The additional mapping of the macropore system makes just a marginal difference. We of course agree that the numerical implementation of the random walk is very simple, but yet it performs satisfyingly compared to the results of the tracer experiments and HYDRUS-1D as we have shown. And yes, it might be indeed the case that macropore systems are not static in space or time but underlie natural variations like changes in worm populations as explained by van Schaik et al. (2014) or anthropogenic influences like plowing. This implies that the macropore domain needs to be re-parameterized. But this problem is not specific to the LAST-Model. Instead, it applies to any kind of soil physical model and we have to deal with this problem if we want to reach real progress.

**R2:** As a curiosity, do you have any idea about why HYDRUS model is able to reproduce site 23 and not site 31?**

AS: In general, you are right. HYDRUS slightly overestimates tracer masses and hence predicts a deeper percolation of water and solutes at site 31. But the shape and course of the simulated profile are still in a pretty good accordance with the observed one, so we would state that HYDRUS is indeed able to moderately reproduce the tracer profile at the end of the experiment on site 31. Obviously, in the HYDRUS model the impact of downward flow determined by the hydraulic conductivity is stronger than the lateral mixing of solutes in the matrix. Hence, because of the higher saturated hydraulic conductivity at site 31, HYDRUS predicts a deeper percolation of water and solutes.

**MINOR COMMENTS:**

**R2**: Please revise carefully the notation, there are still some typos that I had already pointed out in my first revision, e.g.

- In Eq. 2, 3, 4: please use consistent notation for the times sign with the rest of the manuscript

**AS:** Good point, thank you. We will change this.

**R2:** - Eq. 3: is dz = *dzpfd*?

**AS**: As stated below Equation 3, dz is the grid element size in the matrix domain.  $dz_{pfd}$  is the grid element size in the preferential flow domain. They both have different sizes as the two domains also have different extents (dz = 0.1 m;  $dz_{pfd} = 0.05$  m). We will specify this.

**R2:** - Eq. 5: is *dmac* = DM?**

**AS:** Sorry for this error. We have changed DM to *dmac*.

**R2:** - In the text the number of macropore is indicated both with nmac and *nmac**

AS: You are right, thanks. We will use a unified format.

**R2: - Table 1, 2, text: please write consistently f big or fbig**

AS: Thank you. We will revise that.

**R2**: - Eq. 6: It would not be preferable to write dqmix consistently with dzpfd instead of qmix?**

AS: The d in dz and  $dz_{pfd}$  indicates a distance or difference between two depths but qmix is a mixing flux between macropore and matrix. We hence maintain our notation.

**R2:** - Eq. 7: is rM = 2 DM ?**

AS: No,  $r_M$  is the radius of a macropore. As Equation 7 originates from the study of Zehe and Flühler (2001b), we also adopted their notation. To stay consistent with our notation, we will change the variable to *dmac*/2.

**R2**: - Fig. 4: Please use consistent notation in the axes i.e. use the dot and not the comma to separate decimal digits**

**AS:** Thanks, you are right. Figure 4 also originates from the study of Zehe and Flühler (2001b) like Equation 7 (see above). We just adopted their figure and equation in our study. We will update Figure 4 and specify it in the caption.

**R2:** - In my first review, I asked you to highlight the differences/similarities of your model with previous double-continuum models, your replied that "This similarity arises from the fact that both the LAST-Model and double-domain models work with two different domains. But apart from that, the two domains of our model and these of the double-domain models have not much in common because we really established a separate, physically and geometrically described macropore domain with the particle-based Lagrangian approach to simulate water flow and solute transport. The other double-domain models rely on separated overlapping continua..." I would specify that there are already double domain models that do not rely on overlapping continua, but take into account physically and

geometrically separate domains (e.g. Russian, et al. (2013) Water Resour. Res., 49, 8552–8564, doi:10.1002/2013WR014255).

AS: Thank you for this hint. We were not aware of this study. We will check it and implement a respective passage into the discussion section.

**R2**: Considering the large number of parameters that characterise your model I would not introduce new parameters without pointing out which the independent ones are.**

AS: Yes, good point. In general, all observable parameters listed in Table 1 can be freely adjusted in our model and are hence independent from other variables. All other calculated parameters presented in the text like infiltrating or mixing masses are of course dependent on these observable parameters. We will specify this in the caption of Figure 1.

**References**

[revised manuscript text omitted]